# Prediction of source contributions to urban background PM$_{10}$ concentrations in European cities: a case study for an episode in December 2016 using EMEP/MSC-W rv4.15 and LOTOS-EUROS v2.0 - Part.1 The country contributions

Matthieu Pommier[1], Hilde Fagerli[1], Michael Schulz[1], Alvaro Valdebenito[1], Richard Kranenburg[2], Martijn Schaap[2,3]

1 Norwegian Meteorological Institute, Oslo, Norway
2 TNO, PO Box 80015, 3508TA Utrecht, The Netherlands
3 FUB − Free University Berlin, Institut für Meteorologie, Carl-Heinrich-Becker-Weg 6-10, 12165 Berlin, Germany

Correspondence to: matthieu.pommier@met.no

**Abstract.**

A large fraction of the urban population in Europe is exposed to particulate matter levels above the WHO guideline. To make more effective mitigation strategies, it is important to understand the influence on particulate matter (PM) from pollutants emitted in different European nations. In this study, we evaluate a country source contribution forecasting system aimed to assess the domestic and transboundary contributions to PM in major European cities for an episode in December 2016. The system is composed of two models (EMEP/MSC-W rv4.15 and LOTOS-EUROS v2.0) which allows to consider differences in the source attribution.

We also compared the PM$_{10}$ concentrations and both models present satisfactory agreement in the 4-day forecasts of the surface concentrations, since the hourly concentrations can be highly correlated with in-situ observations. The correlation coefficients reach values up to 0.58 for LOTOS-EUROS and 0.50 for EMEP for the urban stations; and 0.58 for LOTOS-EUROS and 0.72 for EMEP for the rural stations. However, the models under-predict the highest hourly concentrations measured by the urban stations (mean underestimation by 36%), which is to be expected given the relatively coarse model resolution used (0.25° longitude × 0.125° latitude).

For the source attribution calculations, LOTOS-EUROS uses a labelling technique, while the EMEP/MSC-W model uses a scenario having reduced anthropogenic emissions and then it is compared to a reference run where no changes are applied. Different percentages (5%, 15% and 50%) in the reduced emissions for the EMEP/MSC-W model were used to test the robustness of the methodology. The impact of the different ways to define the urban area for the studied cities was also investigated (i.e. 1 model grid cell, 9 grid cells and the grid cells covering the definition given by the Global Administrative Area - GADM). We found that the combination of a 15% emission reduction and a larger domain (9 grid cells or GADM) help to preserve the linearity between emission and concentrations changes. The non-linearity, related to the emission reduction

scenario used, is suggested by the nature of the mismatch between the total concentration and the sum of the concentrations

from different calculated sources. Even limited, this non-linearity is observed in the $NO_3^-$, $NH_4^+$ and $H_2O$ concentrations, which

is related to gas-aerosol partitioning of the species. The use of a 15% emission reduction and of a larger city domain also gives

a better agreement in the determination of the main country contributors between both country source calculations.

Over the 34 European cities investigated, $PM_{10}$ was dominated by domestic emissions for the studied episode (December 01st

to 09th 2016). The two models generally agree on the dominant external country-contributor (68% on an hourly-basis) to $PM_{10}$

concentrations. 75% of the hourly predicted $PM_{10}$ concentrations by both models, have the same top 5 main country

contributors. Better agreement is found in the dominant country contributor for primary (emitted) species (70% for POM and

80% for EC) than for the inorganic secondary component of the aerosol (50%), which is predictable due to the conceptual

differences in the source attribution used by both models. The country contribution calculated by the scenario approach

depends on the chemical regime, which largely impacts the secondary components, unlike the calculation using the labelling

approach.

## 1. Introduction.

The adverse health impacts from air pollution and especially from particulate matter (PM) is a well-documented problem (e.g. Keuken et al., 2011; REVIHAAP, 2013; Mukherjee and Agrawal, 2017; Segersson et al., 2017). Furthermore, it affects crop yields (e.g. Crippa et al., 2016), visibility (e.g. Founda et al., 2016) and even the economy (e.g. Meyer and Pagel, 2017). The mass of particulate matter with an aerodynamic diameter lower than 10 µm ($PM_{10}$) is an air quality metric linked to premature mortality at high exposure (e.g. Dockery and Pope, 1994). The World Health Organization (WHO) has established a short-term exposure $PM_{10}$ guideline value of 50 µg/m$^3$ daily mean that should not be exceeded in order to ensure healthy conditions (the long-term exposure guideline is 20 µg/m$^3$ for annual-mean $PM_{10}$) (WHO, 2005). Although policies have been proposed and implemented at the international (e.g. Amann et al., 2011) and national (e.g.D'Elia et al., 2009) levels, European cities still suffer from poor air quality (EEA report 2017), especially due to high $PM_{10}$ concentrations. In short, to further decrease the adverse health impacts of PM in Europe its concentrations need to be reduced further.

$PM_{10}$ concentrations in the atmosphere are highly variable in space and time. Due to the relative short atmospheric life time (from some hours to days), the variability is impacted by local sources, meteorological conditions affecting dispersion and long-range transport as well as chemical regimes controlling the efficiency of secondary formation. $PM_{10}$ consists of both primary and secondary components. Primary $PM_{10}$ components include organic matter (OM), elemental carbon (EC), dust, sea salt (SS) and other compounds. Secondary $PM_{10}$ comprises compounds formed by chemical reactions in the atmosphere from gas-phase precursors. This includes various compounds as nitrate ($NO_3^-$) from nitrogen oxide (NO$_x$) emissions, ammonium ($NH_4^+$) from ammonia (NH$_3$) emissions, sulphate ($SO_4^{2-}$) from sulphur dioxide (SO$_2$) emissions, and a large range of secondary organic aerosol (SOA) compounds from both anthropogenic and biogenic volatile organic compounds (VOCs).

The sources for PM and its precursors are multiple but the main anthropogenic sources are the transport, industries, energy production and agriculture. The main natural sources are composed of forest fires, mineral dust and sea salt. The main sink is the wet deposition. The dry deposition can also be important and depends on the type of land surface such as grass, tree leaves and others; and on meteorological conditions. With these components deriving from various sources, we understand the importance to reflect properly the source contributions in the modelling for policy support.

Many studies have already focused on source receptor relationships to calculate the transport of atmospheric pollutants, with country-to-country relationships (e.g. EMEP Status Report 1/2018) but also over cities (e.g. Thunis et al., 2016; 2018). However, these studies focus on annual means, whereas information is also required on exposure from episodes which cause short-term limit value exceedances throughout Europe. Source apportionment provides valuable information on the attribution of different sources to $PM_{10}$ concentrations. A country source calculation allows to tackle the emissions from the countries responsible for the air pollution episode. Two distinct methodologies have been compared in this study. Indeed, the country source contribution presented hereafter is performed by two regional models, the EMEP/MSC-W model (Simpson et al., 2012) and LOTOS-EUROS (Manders et al., 2017).

The EMEP calculations use reduced anthropogenic emission scenario and compare to a reference run where no changes are applied. It is also known as the scenario approach. With a such simulation comparison, the simulation with reduced emissions over a source region (e.g. a country) allows to highlight the impact of this source on the concentrations over a receptor, hereafter a city. Hence, the scenario approach is useful for analyzing the concentration changes due to emission reductions. On the other hand, one simulation per source is needed to calculate the impact of each source, as done on annual means for each country in each EMEP report (e.g. EMEP Status Report 1/2018). The scenario approach may also lead to a non-linearity in the calculated concentrations, i.e. a slight difference between the concentrations over a receptor and the sum of the estimated concentrations from different sources over this same receptor, as shown by Clappier et al. (2017a). Thus, the scenario approach is more appropriate in the calculation of the source contribution for the primary PM components than for non-linear species such as the secondary components (e.g. Burr and Zhang, 2011, Thunis et al., 2019). LOTOS-EUROS traces the origin of air pollutants throughout a simulation using a labelling approach. The advantage of the labelling technique is the reduction of the computational time, in comparison to the scenario approach. It also quantifies the contribution of an emission source to the concentration of one pollutant at one given location. However, it is not designed to study the impact of emission abatement policies to pollutants concentrations (Grewe et al., 2010; Clappier et al., 2017b) and only traceable atoms can be used in labelling approach, i.e. only conserved atoms (C, N, S), directly related to emission sources, in their different oxidation states. Thus, for example, the origin of ozone ($O_3$) cannot be studied, which can be done with the scenario approach. Even if both methodologies mainly aim to answer two different questions, i.e. the emission control scenarios with the scenario approach and the attribution of concentrations from a source by the labelling technique, it is still useful to estimate the reliability of both methodologies in the estimation of the source contribution to $PM_{10}$ concentrations. For example, it is important to ensure that the non-linearity, related to the perturbation used in the scenario approach, has a limited impact on the calculated contributions and to show that both methodologies may present similar results in the country source attribution.

Both models are part of the operational country source contribution (SC) prediction system for the European cities within the Copernicus Atmosphere Monitoring Service (CAMS). This system aims at attributing country contribution to surface $PM_{10}$ in European cities for 4-day forecasts. The objective of this study is to evaluate the robustness of a new system that provides forecasts of source region resolved PM for European cities. The evaluation of the system is focused on an event occurring between the December 01[st] and 09[th] 2016, which corresponds to the first event listed from the beginning of the development of our system. To do so, the predicted $PM_{10}$ concentrations are compared with observations. The simulations from both models, for the concentrations and the SC calculations are also inter-compared.

Section 2 describes the country SC system composed of the two models and the experiment. Section 3 describes the studied episode and it presents the evaluation of both predictions in terms of $PM_{10}$ concentrations. The methodology used for the SC calculations by both models is explained in Section 4. Then Section 5 gives an overview of the composition and the origin of $PM_{10}$ over the cities predicted by both models, and the issue regarding the non-linearity in the chemistry related to the EMEP

SC calculation. Section 6 is a comparison between the two country SC calculations. Finally, the conclusions are provided in Section 7.

## 2. Description of the country source apportionment system

### 2.1. Overview of the system

Within CAMS, a country SC product has been developed. This is a new forecasting and near-real time source allocation system

for surface $PM_{10}$ concentrations and its different components over all European capitals. The predictions are available online on https://policy.atmosphere.copernicus.eu/SourceContribution.php (last access: 24 January 2020). The concentrations are calculated over the 28 EU capitals plus Bern, Oslo and Reykjavik. Forecasts for Barcelona, Rotterdam and Zurich are also provided. In addition to providing information about the air quality over the selected cities by focusing on $PM_{10}$, this product aims at quantifying the contributions of emissions from different countries in each city (Fig. 1).

The system is composed of predictions from two regional models (the EMEP/MSC-W model and LOTOS-EUROS), using two distinct source contribution calculation methodologies. The EMEP/MSC-W chemistry transport model (Simpson et al., 2012) has been used for decades to calculate source receptor relationships between European countries (including Russia) (e.g. EMEP Status Report 1/2018) and the LOTOS-EUROS chemistry transport model (Manders et al., 2017) has also been used in several source apportionment studies over Europe, especially for PM (Hendriks et. al., 2013; 2016; Schaap et al., 2013). Both

models are involved in the operational air quality analysis and forecasting for Europe in the CAMS regional ensemble system (Marécal et al., 2015) and for China (Brasseur et al., 2019). For the simplicity of the reading, the EMEP/MSC-W model is hereafter referred to as EMEP model.

Both models are Eulerian models but there are differences between these two models such as the calculation of the planetary boundary layer (PBL) and of the advection, the vertical resolution. There are also differences in the presence of the secondary

organic aerosol (included in the EMEP model and not in LOTOS-EUROS), $PM_{10}$ diagnosing particle water explicitly in the EMEP model and not in LOTOS-EUROS, the calculation of the biogenic emissions, the description of the gas-phase chemistry and the treatment of dust (from agriculture and traffic are included in LOTOS-EUROS and not in the EMEP model).

The main details about the models and the experiment are provided in the Table 1 and a more complete description is provided in the following Sections.

### 2.2. Description of the EMEP model

The EMEP model is a 3-D Eulerian chemistry-transport model described in detail in Simpson et al. (2012). Initially, the model has been aimed at European simulations, but the model has also been used over other regions and at global scale for many years (e.g. Jonson et al., 2010). The EMEP model version rv4.15 has been used here in the forecast mode. The version rv4.15 has been described in Simpson et al. (2017) and references cited therein. The main updates since Simpson et al. (2012) and

used in this work, concern a new calculation of aerosol surface area (now based upon the semi-empirical scheme of Gerber,

1985), revised parameterizations of $N_2O_5$ hydrolysis on aerosols, additional gas-aerosol loss processes for $O_3$, $HNO_3$ and $HO_2$,

a new scheme for ship $NO_x$ emissions, a new calculated natural marine emissions of dimethyl sulphid (DMS), and the use of

a new land-cover (used to calculate biogenic VOC emissions and the dry deposition) (Simpson et al., 2017). This version is

the official EMEP Open Source version that was released in September 2017 (Tab. 1)

Vertically, the model uses 20 levels defined as sigma coordinates (Simpson et al., 2012). The PBL is located within

approximately the 10 lowest model levels (~5 levels below 500 m), and the top of the model domain is at 100 hPa. The PBL

height is calculated, based on the turbulent diffusivity coefficient as described in the EMEP Status Report (2003). The

numerical solution of the advection terms is based upon the scheme of Bott (1989).

The chemical scheme couples the sulphur and nitrogen chemistry to the photochemistry using about 140 reactions between 70

species (Andersson-Sköld and Simpson, 1999; Simpson et al. 2012). The chemical mechanism is based on the "EMEP scheme"

described in Simpson et al. (2012) and references therein.

The biogenic emissions of isoprene and monoterpene are calculated in the model by emission factors as a function of

temperature and solar radiation (Simpson et al., 2012). The soil-NO emissions of seminatural ecosystems are specified as a

function of the N-deposition and temperature (Simpson et al., 2012). The biogenic DMS emissions are calculated dynamically

during the model calculation and vary with the meteorological conditions (Simpson et al., 2016).

PM emissions are split into EC, OM (here assumed inert) and the rest of primary PM defined as the remainder, for both fine

and coarse PM. The OM emissions are further divided into fossil-fuel and wood-burning compounds for each source sector.

As in Bergström et al. (2012), the OM/OC ratios of emissions by mass are assumed to be 1.3 for fossil-fuel sources and 1.7

for wood-burning sources. The model also calculates windblown dust emissions from soil erosion. Secondary aerosol consists

of inorganic sulphate, nitrate and ammonium, and SOA; the latter is generated from both anthropogenic and biogenic

emissions, using the 'VBS' scheme detailed in Bergström et al (2012) and Simpson et al. (2012).

The main loss process for particles is wet-deposition, and the model calculates in-cloud and sub-cloud scavenging of gases

and particles as detailed in Simpson et al. (2012). Wet scavenging is treated with simple scavenging ratios, taking into account

in-cloud and sub-cloud processes.

In the EMEP model, the 3D precipitation is needed. An estimation of this 3D precipitation can be calculated by EMEP if this

parameter is missing in the meteorological fields as in the data used in this work (see Section 2.4). This estimate is derived

from large scale precipitation and convective precipitation. The height of the precipitation is derived from the cloud water.

Then, it is defined as the highest altitude above the lowest level, where the cloud water is larger than a threshold taken as

$1.0 \times 10^{-7}$ kg water per kg air. Precipitations are only defined in areas where surface precipitations occur. The intensity of the

precipitation is assumed constant over all heights where they are non-zero

Gas and particle species are also removed from the atmosphere by dry deposition. This dry deposition parameterization follows standard resistance-formulations, accounting for diffusion, impaction, interception, and sedimentation.

## 2.3. Description of LOTOS-EUROS

The LOTOS-EUROS model is an off-line Eulerian chemistry-transport model which simulates air pollution concentrations in the lower troposphere solving the advection-diffusion equation on a regular latitude-longitude-grid with variable resolution over Europe (Manders et al., 2017) (Tab. 1).

The vertical grid is based on terrain following vertical coordinates and extends to 5 km above sea level. The model uses a dynamic mixing layer approach to determine the vertical structure, meaning that the vertical layers vary in space and time. The layer on top of a 25 m surface layer follows the mixing layer height, which is obtained from the European Centre for Medium-Range Weather Forecasts (ECMWF) meteorological input data that is used to force the model. The horizontal advection of pollutants is calculated applying a monotonic advection scheme developed by Walcek and Aleksic (1998).

Gas-phase chemistry is simulated using the TNO CBM-IV scheme, which is a condensed version of the original scheme (Whitten et al, 1980). Hydrolysis of $N_2O_5$ is explicitly described following Schaap et al. (2004).

LOTOS-EUROS explicitly accounts for cloud chemistry computing sulphate formation as a function of cloud liquid water content and cloud droplet pH as described in Banzhaf et al. (2012). For aerosol chemistry the thermodynamic equilibrium module ISORROPIA2 is used (Fountoukis and Nenes, 2007).

The biogenic emission routine is based on detailed information on tree species over Europe (Schaap et al., 2009). The emission algorithm is described in Schaap et al. (2009) and is very similar to the simultaneously developed routine by Steinbrecher et al. (2009). Dust emissions from soil erosion, agricultural activities and resuspension of particles from traffic are included following Schaap et al. (2009).

As in the EMEP model, the 3D precipitation is needed and cloud liquid water profiles are used to diagnose cloud base height and where below and incloud scavenging takes place. The wet deposition module accounts for droplet saturation following Banzhaf et al. (2012). Dry Deposition fluxes are calculated using the resistance approach as implemented in the DEPAC (DEPosition of Acidifying Compounds) module (van Zanten et al., 2011). Furthermore, a compensation point approach for $NH_3$ is included in the dry deposition module (Wichink Kruit et al., 2012).

## 2.4. Description of the experiment

The study focuses on the period from December 01st to 09th 2016. In our system, the forecasts provided by the EMEP model cover a slightly different regional domain than LOTOS-EUROS (Tab. 1). To perform properly the analysis between both models, we have harmonized the use of different parameters such as the horizontal resolution, the anthropogenic emissions used, the definition of the city area and meteorological data used (Tab. 1). This harmonization has been revealed important for

such comparison and increases the consistency of the model results. The impact of such choices is illustrated for the city definitions, for which subjective choices can be made causing inconsistencies.

An initial spin-up of 10 days was conducted. Both models provide four-day air quality forecasts, and the simulations have been defined as "forecast-cycling experiments", i.e. the predicted fields have been used to initialize successive four-day forecasts (e.g Morcrette et al., 2009). The pollution transport in both models is based on forecasted meteorological fields at 12 UTC from the previous day, with a 3-hour resolution, calculated by the Integrated Forecasting System (IFS) of ECMWF. These forecasted meteorological fields correspond to the fields which were used in the online SC production for these dates. The ECMWF operational system does not archive 3D precipitation forecasts, which is needed by the EMEP model and LOTOS-EUROS as mentioned in Sections 2.2 and 2.3. Therefore, a 3D precipitation estimate is derived from IFS surface variables (large scale and convective precipitations) in the EMEP model and the 3D field is based on the cloud liquid water profile in LOTOS-EUROS.

The boundary conditions (BCs) at 00UTC of the current day from the atmospheric Composition module (C-IFS) have been used. These BCs are specified for ozone ($O_3$), carbon monoxide (CO), nitrogen oxides (NO and $NO_2$), methane ($CH_4$), nitric acid ($HNO_3$), peroxy-acetyl nitrate (PAN), $SO_2$, ISOP, ethane ($C_2H_6$), some VOCs, sea salt, Saharan dust and $SO_4$. In LOTOS-EUROS, sea salt BCs have not been used as these are shown to be overestimated in comparison with the model. In the EMEP model, the sea salt parameter has been used. This may cause a difference between both models in the estimation of the contribution from sea salt especially for the coastal cities.

Both models use the TNO-MACC emission data set for 2011 on $0.25° \times 0.125°$ (longitude-latitude) resolution (Kuenen et al., 2014, see http://drdsi.jrc.ec.europa.eu/dataset/tno-macc-iii-european-anthropogenic-emissions) and the forest fire emissions are from GFASv1.2 inventory (Kaiser et al., 2012).

Since the study aims to quantify the contributions of long-range transport in each city to the urban background $PM_{10}$, the effect of the choice of the receptor, i.e. the city domain, has been tested. The city receptor has been defined by three definitions: 1 grid cell (i.e. $0.25°$ lon $\times 0.125°$ lat, corresponding to the emissions data set resolution), 9 grid cells and the all the grid cells covering the administrative area provided by the database of Global Administrative Areas (GADM, https://gadm.org/data.html). The latter is the more precise definition in terms of build-up area, however it may represent a large region for a definition of a city as shown in Fig. S1 (e.g. London, Nicosia, Riga, Sofia). It is important to explain that this study does not aim to quantify the contribution to $PM_{10}$ at a street scale as done in Kiesewetter et al. (2015) but over the full area defining the cities. The relatively coarse definition of the cities is comparable to the definition used in previous studies as in Thunis et al. (2016) who used an area of $35 \times 35$ km$^2$ or in Skyllakou et al. (2014) who used a radius of 50 km from the city center.

For the contribution, we also have harmonized the definition of the natural contributions. The natural contributions are defined in this study as the sum of the contributions from sea salt, dust and forest fires, except for the BCs. In LOTOS-EUROS, the natural sources (e.g. dust) coming from the boundaries are classified as BCs and not natural.

## 3. Evaluation of the predicted surface concentrations during the episode

During December 2016, a PM episode with medium intensity (no more than three consecutive days beyond the WHO PM$_{10}$ threshold) developed across North-Western Europe. As a consequence of a high pressure system over central Europe pollutants concentrations were built up over western Europe (see http://policy.atmosphere.copernicus.eu/reports/CAMSReportDec2016-episode.pdf last access: 24 January 2020).

From December 1$^{st}$ to 2$^{nd}$, high concentrations were measured and predicted over Paris (Figures 1 & 2). In Figure 2, we can also see from December 3$^{rd}$ to December 8$^{th}$, that levels of PM$_{10}$ were elevated in Western Europe. Especially on December 6$^{th}$ and 7$^{th}$, concentrations at some measurement stations in France, Belgium, the Netherlands, Germany and Poland, exceeded the daily limit value of 50 μg/m$^3$ (e.g Fig. S2 – see Section 3.2 for more details about the observations).

During the following days relatively stable conditions with slow southerly winds characterized the episode until fronts moved in western Europe at the 9$^{th}$. Large concentrations (>60 μg/m$^3$) were also predicted between December 6$^{th}$ and 9$^{th}$ over the Po Valley and over UK on December 6$^{th}$ (Figs. 2 and S2).

### 3.1. Statistical metrics used

To properly estimate the quality of these forecasts, five statistical parameters have been used, such as the Pearson correlation (r), the Mean Bias (MB), the Normalized Mean Bias (NMB), the Root-Mean-Square Error (RMSE) and the Fractional Gross Error (FGE). The ideal score of these parameters is 0 except for the correlation which is 1.

The MB provides the information about the absolute bias of the model, with negative values indicating underestimation and positive values indicating overestimation by the model. The NMB represents the model bias relative to the reference. The RMSE considers error compensation due to opposite sign differences and encapsulates the average error produced by the model. The FGE is a measure of model error, ranging between 0 and 2 and behaves symmetrically with respect to under- and overestimation, without over emphasizing outliers.

We have used M and R as notation to refer, respectively, to model and the reference data (e.g. observations), and N is the number of the reference data set (e.g. number of observations).

Thus, MB is calculated by equation (1) and expressed in μg/m$^3$:

$$MB = \frac{\sum_{i=1}^{N}(M_i - R_i)}{N} \qquad (1)$$

NMB is calculated by equation (2):

$$\text{NMB} = \frac{\sum_{i=1}^{N}(M_i - R_i)}{\sum_{i=1}^{N} R_i} \times 100\% \qquad (2)$$

RMSE is calculated by equation (3) and expressed in µg/m³:

$$\text{RMSE} = \sqrt{\frac{\sum_{i=1}^{N}(M_i - R_i)^2}{N}} \qquad (3)$$

and FGE is calculated by equation (4) and dimensionless:

$$\text{FGE} = \frac{2}{N} \sum_{i=1}^{N} \frac{|M_i - R_i|}{|M_i + R_i|} \qquad (4)$$

### 3.2. Comparison with observations

### 3.2.1. Methodology

In order to evaluate the reliability of the predictions over each city, the modelled hourly PM$_{10}$ concentrations have been compared with the AirBase data (see https://acm.eionet.europa.eu/databases/airbase/). The traffic stations were not included

in the comparison since a regional model with a somewhat coarse resolution will not be able to calculate very large concentrations (e.g. hourly concentration higher than 200 µg/m³) which may be measured by these stations. Indeed, the concentrations calculated by a regional model over cities are mostly representative of the urban background. By knowing this point, we have stated that a comparison with the observations presenting for example a correlation coefficient equal to 0.5 or NMB lower than 15% are reasonable results (r ≥ 0.7 and NMB ≤ 10% are good results). The observations have also been

categorized into two sets of data by differentiating the rural stations to the urban stations (as shown in Fig. S2). This follows the procedure done in the yearly evaluation of the EMEP model over Europe (e.g. EMEP Status Report 1/2018). Due to the relatively coarse definition of a city, it appears that stations classified as rural may be present in our city domain.

It was noticed that for the smaller definition of the city edges, i.e. 1 grid cell, there were no rural stations within the city domain. Obviously, by increasing the size of the city domain, to 9 grid cells or by using the GADM definition, the number of rural

stations present within the city domain increases. Indeed, all the hourly measurements are averaged within the city boundary, by separating the urban and the rural stations. A comparison with these two types of stations can highlight a difference between the urban background and the urban concentrations. For such comparison, the model concentrations are also averaged over the city domain.

### 3.2.2. Results

Figures 3 and 4 show the comparison between the hourly averaged observations within the city edges defined by the 9 grid cells definition, and the predictions from EMEP and from LOTOS-EUROS respectively.

Figures 3 and 4 show that for the urban stations, the different predictions from a same model, for the same date, are consistent since the values for the statistical parameters are relatively constant. It is noticed; however, that the bias is slightly reduced when the starting date of the forecast is closer to the target date. The available observations and thus the stations may also

differ from day to day (e.g. Fig. S2a). Figures 3 and 4 also show that despite many differences, the models have very similar performances in comparison with the urban stations.

In Figure 3, it is also clear that the EMEP model has difficulties to reproduce the highest concentrations measured by the urban stations which are probably smoothed by the model over the large grid cells as the ones defining the cities. The underestimation in the largest urban concentrations is highlighted by the comparison with the rural stations. This also shows that over the area

defining the cities there is a large variability in the measured $PM_{10}$ concentrations and that few stations are not necessarily representative of the model grids. It also shows with such resolution; the model represents urban background concentrations. Only 5 cities have measurements defined as rural stations by using the 9 grids definition (i.e. Amsterdam, Berlin, Luxembourg, Rotterdam and Vienna) while there are up to 19 cities with urban stations. By comparing only the 5 cities having urban and rural stations, the agreement between EMEP and the urban stations is largely improved as shown in Fig. S3. We can also notice

that the difference in concentrations predicted by the EMEP model between both types of stations is also reduced. This shows that for these five cities, the predicted $PM_{10}$ concentrations on December $6^{th}$ are higher than over the other cities.

LOTOS-EUROS is less correlated with the concentrations measured by the rural stations than EMEP (Fig. 4). However, as EMEP, LOTOS-EUROS also presents a lower bias with these rural stations in comparison with the urban stations. This is predictable since with such resolution, the model calculates mainly the urban background concentrations. By comparing the 5

cities having urban and rural stations, as done with EMEP, only the bias and the FGE between the predictions and the urban measurements are improved (Fig S4). It is also worth noting that the concentrations predicted by LOTOS-EUROS over these 5 cities are lower than the ones calculated by the EMEP model (in Fig. S3).

By using the GADM definition, the number of cities having rural stations decreases to 2 while the number of cities with the urban stations remains identical.

In general, both models present similar performance with the observations especially for the NMB, RMSE and FGE as presented in Figures S5 and S6. These figures show an overview of the statistical parameters for all 4-d forecasts, i.e. the dates from December $01^{st}$ to $12^{th}$ 2016 with a starting date from December $01^{st}$ to $09^{th}$, for all the cities defined by 9 grid cells, in comparison with the concentrations measured at the urban and the rural stations, respectively.

As already shown by Figs. 3 and 4, LOTOS-EUROS shows slightly better correlation coefficients with the urban stations than

EMEP (Fig. S5, in average $R_{LOTOS-EUROS}=0.31$, $R_{EMEP}=0.25$; with a maximum of 0.58 for LOTOS-EUROS and 0.5 for EMEP) and EMEP presents better correlations with the few rural stations (Fig. S6, in average $R_{LOTOS-EUROS}=0.23$, $R_{EMEP}=0.35$; with a maximum of 0.58 for LOTOS-EUROS and 0.72 for EMEP). However, the limited number of cities having rural stations explain the larger variability in the correlations compared to the correlations found with the urban stations. Similar results are found by using the GADM definition (not shown) while by using only 1 grid to define the city edges, the correlation

coefficients with the urban stations are larger (up to 0.8), with an increase in the bias and a decrease in the RMSE (Fig. S7).

In average, both models have a FGE equal to 0.5 over the cities defined by 9 grid cells with the urban stations and 0.4 with the rural stations. For the RMSE, it is 33 µg/m$^3$ with the urban stations and 11 µg/m$^3$ with the rural stations. While both models underestimate the PM$_{10}$ concentrations by 36% in average by using the urban sites, EMEP overestimates by 6% with the rural stations and LOTOS-EUROS underestimates by 6%.

Performances of both models are improved with daily means, especially with better correlation coefficients (not shown). For example, with the cities defined by 9 grid cells, the correlation coefficients reach 0.8 with the urban stations for EMEP and LOTOS-EUROS and 0.98 with the rural stations for EMEP. However, a lot of negative correlation coefficients between LOTOS-EUROS and the rural stations are noticed. The correlation coefficient with the rural stations remains difficult to interpret related to the limited number of stations available. Thus, EMEP presents a mean correlation coefficient equal to 0.4

with the urban and rural stations, and LOTOS-EUROS has a mean correlation of 0.5 with the urban stations and only 0.06 with the rural stations. Better scores with the FGE and the RMSE are also noticed in comparison to the hourly evaluation (not shown). Both models present with these 9 grids definition a mean FGE of 0.5 with the urban stations and 0.3 for the rural stations and a mean RMSE of 21 µg/m$^3$ with the urban stations and 10 µg/m$^3$ with the rural stations.

**3.3. Inter-comparison in the concentrations predicted by both models**

The second analysis has been focused on the agreement between both models. During the episode, all 4-d forecasts present a high correlation between the PM$_{10}$ predicted by the EMEP model and LOTOS-EUROS as shown by Figure 5a. These correlations vary from day to day and city by city but remain large for the different simulated periods (median = 0.7).

There is no clear geographical pattern in terms of performance between the two models, even if the central European cities (e.g. Budapest, Vienna, Warsaw) presented the larger differences (Fig. 5b). These differences may be explained by slightly

lower Secondary Inorganic Aerosols (SIA = $NO_3^- + NH_4^+ + SO_4^{2-}$) in LOTOS-EUROS for these cities but also by lack of water in LOTOS-EUROS (which is not diagnosed as mentioned in Sect. 2). Moreover, it confirms the larger PM$_{10}$ concentrations predicted by EMEP than by LOTOS-EUROS for the five cities plotted in Figs. S3 and S4. It is also worth noting that LOTOS-EUROS predicts more sea salt and dust for almost all the cities during the studied period (Fig. S8) which is representative of the overall feature over the regional domain (not shown). Actually, it was noticed that for the predicted PM$_{10}$ with the larger

positive NMB (EMEP predicting larger PM$_{10}$ concentrations), EMEP has more SIA and "other" than LOTOS-EUROS (Figure S9a), while the PM$_{10}$ from LOTOS-EUROS is dominated by natural components when a larger negative NMB is predicted (Figure. S9b).

**4. Methodology of the source contribution calculation**

**4.1 The EMEP model**

**4.1.1 Emission reductions**

The SC calculation follows the methodology uses in each EMEP annual report to quantify the annual country-to-country source receptor relationships (e.g. EMEP Status Report 1/2018). The experiment is based on a reference run, where all the anthropogenic emissions are included. The other runs are the perturbation runs. These runs correspond to the simulations where the emissions from every considered country are reduced by 15%. As explained in Wind et al. (2004), a reduction by 15% is

sufficient to give a clear signal in the pollution changes. It also gives a negligible effect from non-linearity in the chemistry even if in this work it has been estimated.

The perturbation runs are done for anthropogenic emissions of CO, $SO_x$, $NO_x$, $NH_3$, NMVOC and PPM (primary particulate matter). For computational efficiency, in the perturbation calculations, all anthropogenic emissions in the perturbation runs have been reduced here simultaneously. This simultaneous reduction differs from the methodology uses in each EMEP annual

report where the emissions are reduced individually.

There are in total 31 runs for each date with reduced anthropogenic emissions. Each run corresponds to the perturbations for one of the 28 countries related to the 28 EU capitals, plus Iceland, Norway and Switzerland, giving the contribution for each country.

To calculate the concentration of the pollutant integrated over the studied area, i.e. a selected city, coming from a source, we

follow the equation (5):

$$C_{source} = \frac{C_{reference} - C_{pertubation}}{x} \qquad (5)$$

With x the reduction in % (i.e. 0.15), $C_{reference}$ is the concentration of the pollutant integrated over the studied area from the reference run and $C_{pertubation}$ is the concentration of the pollutant integrated over the studied area from the perturbation run. Thus, by differentiating over the studied area, the concentration from the perturbated run with the concentration provided by the reference run, we have an estimation of the influence of the source (i.e. country). By scaling with the reduction used

(parameter x), it gives the estimated concentration related to the source.

**4.1.2 Issue concerning the chemical non-linearity**

The reason why emissions should not be perturbed by 100% in the model simulations is to stay within the linear regime of involved chemistry. Even limited, such methodology may still introduce a non-linearity in the chemistry. The total $PM_{10}$ over the receptor should be identical theoretically to the sum of the $PM_{10}$ originated from the different sources. This is not always

the case and the difference between the total $PM_{10}$ and the sum from the various sources may lead to negative or positive concentrations. This is a result of the perturbation used which is assumed to be linear to a 100% perturbation.

The 15% emission reduction has been used during many years for the annual country-to-country source receptor relationships calculations (e.g. EMEP Status Report 1/2018). Clappier et al. (2017a) have already shown the robustness of the methodology

at the country scale on yearly averages and for the highest daily concentrations. However, this emission reduction was not

used for smaller areas. Thus this 15% emission reduction for the study over a city and on hourly basis has been tested, in order

to assess the robustness of the calculations. 5% and 50% were the other selected emission reductions. In total, 847 4-day runs

have been performed in this work (9 reference runs, and 9 dates × 31 countries × 3 perturbations runs).

Furthermore, by reducing the emissions simultaneously or separately may lead to a different result in the concentrations, but

as mentioned previously, this effect is not addressed in this work for computational reason.

**4.2. LOTOS-EUROS**

A labelling technique has been developed within each LOTOS-EUROS simulation (Kranenburg et al., 2013). An important

advantage of the labelling technique is the reduction of computation costs and analysis work associated with the calculations.

The source apportionment technique has been previously used to investigate the origin of PM (Hendriks et al., 2013; 2016),

$NO_2$ (Schaap et al., 2013), and nitrogen deposition (Schaap et al., 2018).

Besides the concentrations of all species, the contributions of a number of sources to all components are calculated. The

labelling routine is only implemented for primary, inert aerosol tracers and chemically active tracers containing a C, N (reduced

and oxidized) or S atom, as these are conserved and traceable. This technique is therefore not suitable to investigate the origin

of e.g. $O_3$ and $H_2O_2$, as they do not contain a traceable atom. The source apportionment module for LOTOS-EUROS provides

a source attribution valid for current atmospheric conditions as all chemical conversions occur under the same oxidant levels.

For details and validation of this source apportionment module we refer to Kranenburg et al. (2013).

To avoid violating the memory size and avoid excessive computation times it was chosen to trace the EU-28 countries,

supplemented by Norway and Switzerland. For convenience, a number of small countries was combined with a neighboring

state. For example, Switzerland and Liechtenstein as well as Luxembourg and Belgium were combined. In addition, all sea

areas were combined into one source area. To be mass consistent, all non-specified regions, natural emissions and as well as

the combined impact of initial conditions and boundary conditions were given labels as well.

**5. Information provided by the Source Contribution calculations**

**5.1 In the EMEP calculations**

As presented in Fig. 1, the country contributions to the predicted $PM_{10}$ concentrations in the cities is provided in our products.

Figure 6 presents the mean composition for the "Domestic", "30 European countries" and "Others" $PM_{10}$ contributions for all

cities, for all 4-d predictions and split into negative and positive concentrations. This figure is a result of the perturbation runs

by separating the positive and the negative concentrations obtained in the calculations. The concentrations have also been

gathered by their calculated origin. The "Domestic" contribution corresponds to the contribution from the domestic country to

the city (for example from France to Paris). The "30 European countries" corresponds to the other 30 European countries used

in the study. "Others" gathers mainly natural sources, the other European countries included in the regional domain (and not included in our SC calculations, e.g. Turkey) and the boundary conditions. This figure gives a graphical illustration of the composition of the different contributions and presents the effect of the non-linearity. Indeed, the positive concentrations shows the overall composition for each contribution, while the chemical reason of the non-linearity is highlighted by the negative contribution to the predicted $PM_{10}$ concentrations.

The main contributors to the "Domestic" $PM_{10}$ are POM (~20%) and rest PPM (~30%) (which corresponds to the remainder of coarse and fine PPM), as noticed for the positive concentrations (Fig. 6a). Actually, the variation in the mean concentrations is mainly influenced by the variation in these primary components. $NO_3^-$ is also an important component of these "Domestic" $PM_{10}$. The value of the mean concentration depends on the city definition and so on the average of the concentrations over different size of city. The mean $PM_{10}$ concentration over a smaller area is larger showing that with a smaller grid, the $PM_{10}$ is less diffused over the integrated area. The "30 European countries" $PM_{10}$ is mainly influenced by $NO_3^-$ (by 38%) (Fig. 6b).

Overall, 45% of the contributions to the $PM_{10}$ calculated over the selected cities for this episode are "Domestic" and essentially due to primary components. 35% are from the "30 European countries", essentially $NO_3^-$ and 25% are from "Others" mainly composed of natural sources (representing 50% of "Others"). Obviously, this feature is an overview of all selected cities for all the studied dates and it can vary from city to city and from date to date.

By comparing the $PM_{10}$ concentrations calculated over the same city edges but by using different percentages in the perturbation runs, we have calculated the impact of the non-linearity for each contribution and presented in Figure 7. This non-linearity has been calculated for each hourly concentration as the standard deviation of the hourly contribution (which can be positive or negative) obtained by the three reduced emissions scenarios and weighted by the hourly total concentration by following the equation (6):

$$NONLIN_{Contrib} = \frac{\sqrt{\frac{\sum_{i=1}^{n}\left(Ccontrib_i - \overline{Ccontrib}\right)^2}{n}}}{Ctot} \times 100\% \qquad (6)$$

n corresponds to the number of perturbations used (n=3), Ccontrib is the hourly $PM_{10}$ concentration for a specific contribution ("Domestic" or "30 European countries" or "Others") and Ctot is the hourly $PM_{10}$ concentration. This mean non-linearity due to the "Domestic" contribution represents in maximum 0.9% of the total $PM_{10}$. This non-linearity from the "30 European countries" contribution, counts for 0.7% of the total $PM_{10}$ and 1.5% from "Others". Actually, the non-linearity from the "Others" depends on the non-linearity from the two other contributions. The mean non-linearity is not homogenously distributed over all cities as shown in Figure S10 and may vary from date to date (not shown). It has remained limited even if some hourly contributions show higher non-linearity. In maximum, 3% of the calculated hourly contributions for all 4-day forecasts over the selected cities have a non-linearity higher than 5% (not shown). This shows that due to the methodology used in the EMEP model, based on a reduced emission scenario, the non-linearity in the chemistry has a limited impact on the

SC calculation. This non-linearity is slightly reduced by using the larger domains to define the cities (e.g. 9 grids) (Fig. 7). This also shows that the responses to perturbation runs are robust, even if only the non-linearity in the chemistry related the

perturbation used, and not the one related to the reduction of each emission precursor, has been estimated in this study as mentioned in Section 4.1.

Negligible negative contributions have been calculated for the "Domestic" and "30 European countries" contributions (Figs. 6a & b) and small negative contributions are predicted in "Others" (Fig. 6c). These negative $PM_{10}$ are a result of negative values in $NO_3^-$, $NH_4^+$ and $H_2O$ which is a consequence of gas-aerosol partitioning of the species. Indeed, $NH_3$ reacts with nitric

acid ($HNO_3$) to form ammonium nitrate ($NH_4NO_3$). This is an equilibrium reaction, and thus the transition from solid to gaseous phase depend on relative humidity (e.g. Fagerli and Ass, 2008; Pakkanen, 1996). This shows that, for example, a reduction in $NO_x$ over a country which impacts the selected city, does not necessarily only impact the $NO_3^-$ over this city, but may also have an effect on $NH_3$ chemistry over a second region. This second region may also have itself an impact on the selected city. This combination of $NO_x$ and $NH_3$ chemistry from different regions may lead at the end to these negative concentrations.

The impact of the percentage used in the perturbation runs and the size of the city edges have no significant impact in the amount of negative "Others" $PM_{10}$ concentrations. The impact of both parameters is more visible on the "Domestic" and "Rest of Europe" concentrations but it remains very small.

Averaging out over the larger grids reduces globally the non-linearity. The 15% emission reduction also reduces the negative non-linearity in the "Domestic" concentrations (e.g. $H_2O$ for the 9 grids and GADM runs).

**5.2 In the LOTOS-EUROS calculations**

As presented with the EMEP predictions, Figure 8 presents the mean composition for the "Domestic", "30 European countries" and "Others" $PM_{10}$ contributions for all cities, for all 4-d predictions provided by LOTOS-EUROS. The definition of "Others" is slightly different than the EMEP one since e.g. the dust from agriculture and traffic is included (see Sect. 2). For an easier comparison, the result for the EMEP model using the 15% emission reduction has also been plotted with thinner charts, even

if, as just mentioned, the definition of "Others" slightly differs between both models.

First of all, during the episode, LOTOS-EUROS confirms the general trend calculated by the EMEP model, i.e. the dominant contribution to the surface $PM_{10}$ is "Domestic", ranging between 40% and 48% of the predicted $PM_{10}$ over all selected cities and for all the studied dates. However, LOTOS-EUROS always presents more "Domestic" $PM_{10}$ than the EMEP model. LOTOS-EUROS also predicted slightly more influence from "Others" than the "30 European countries" with a ratio close to

25-30% each. As reminder, the EMEP model predicted a slightly larger influence from the "30 European countries" (35%) than from "Others" (25%).

As with the EMEP model, the mean $PM_{10}$ concentration over the smaller city definition is larger and the "Domestic" $PM_{10}$ is largely driven by POM. In the list of LOTOS-EUROS $PM_{10}$ components there is one named "Rest". "Rest" corresponds to the

difference between the total PM$_{10}$ and the sum of all the components, and Fig. 8 shows that it is also a large component of this "Domestic" PM$_{10}$. POM and "Rest" represent each between 25% and 30% of these "Domestic" PM$_{10}$.

The large influence of $NO_3^-$ (48%) in the "30 European countries" PM$_{10}$ is also calculated by LOTOS-EUROS, as well as the large contribution of the natural components (60%) in "Others". It is noteworthy to see that, even small, the dust emitted by the road traffic and the agriculture is not negligible in these "Others" PM$_{10}$ (~10%).

## 6. Comparison between both country source contribution calculations

Section 3 has highlighted the similar performance from both models in the prediction of the PM$_{10}$ concentrations over the European cities with observations. It has also been shown in Section 3 that both models are representative for a large area and the predictions can underestimate the concentrations and the contributions for the larger concentrations measured by a specific station. Section 5 has shown similar results in terms of composition of these PM$_{10}$. It is also noteworthy to see in Figure 9 that both SC calculations present a high rate of agreement over the selected period with the common simulated components and the PM$_{10}$ calculated by both models. This rate corresponds to the number of occurrences in the dominant contributor calculated for each hourly concentration in the 4-day forecast over each city. So, a number as 100% over a city shows that both models predict the same dominant country contributor during a 4-day forecast. In Fig. 9, both models show that, by using the 9 grid cells definition, in average 68% of the hourly predicted PM$_{10}$ concentrations have the same dominant country contributor. In average, 50% of the secondary inorganic aerosols predicted by both models over all the cities and all 4-day forecasts have the same main contributor. This value goes up to 70% for POM and 80% for EC. For the two primary components (POM and EC) the median is larger, with a value of 77% and 93% respectively, showing that the mean value in the agreement for both compounds is reduced by a few low values (Fig. 9). On a daily basis, the mean agreement is slightly improved, e.g. 70% of agreement for the PM$_{10}$ (Fig. S11). The main improvement is calculated for EC, with a median equal to 100% (Fig. S11).

The lower agreement for the SIA is predictable due to the various origins (chemistry and primary emissions) for these particulates and the different aerosols treatment (gas-aerosols partitioning) in both models. It is also related to the differences in both methodologies (e.g. Clappier et al, 2017b). Indeed, an emission reduction and a labelling technique will not necessarily provide the same results for the secondary PM. An emission reduction depends on the atmospheric composition already present. For example, an amount of NO$_x$ emitted over a source can result in a certain NH$_4$NO$_3$ concentration in the receptor. If this NO$_x$ is emitted in excess (NH$_3$ limited regime), a NO$_x$ emission reduction will have a small effect at the receptor point. On the other hand, in the NO$_x$ limited regime, the same NO$_x$ reduction will have a large impact. The labelling method will give the same result in both cases while the scenario approach will give different results.

This agreement varies from city to city (Fig. 10) but it is shown, in addition to the example of PM$_{10}$ (Fig. 5), that central European cities often present a limited agreement due to their central location and the influence of various countries. This limited agreement is also sometimes observable for the cities close to the edge of the regional domain (Fig. 10), which could

be explained by the influence of the boundary conditions as the dust transported from other regions (e.g. Valetta influenced by dust from Sahara).

The mean agreement increases up to 75% for determination in the top 5 of the main country contributors to $PM_{10}$ (Fig 11). In that case, the rate is calculated for the five main country contributors. A score of 100% means both models predict the same five main country contributors for each hourly concentration, but not necessarily in the same order. This rate is around 70%

for $SO_4^{2-}$, EC and POM and close to 60% for $NO_3^-$ and equal to 65% for $NH_4^+$ (Fig. 11). As for the dominant country contributor, the agreement is slightly improved by using daily means, e.g. we found 76% of agreement with the $PM_{10}$ (not shown).

It is also important to notice that these overall agreements are neither significantly influenced by the definition of the cities area nor on the perturbation percentage tested for the EMEP SC calculations (Fig. S12). The agreement is slightly better by

using the smaller area (1 grid) in the determination of the dominant country contributor and slightly better by using a large domain (9 grids or GADM) in the determination of the 2 and 5 main contributors.

Overall, a perturbation run using a reduction of 15% and the use of a larger city area (e.g. GADM or 9 grids) allow a better determination in the country contributors, with a better agreement with LOTOS-EUROS and limit the impact of the non-linearity in the chemistry.

**7. Conclusions**

By focusing on a specific event, occurring from December 01st to 09th 2016 over Europe, this work is the first attempt to evaluate the source contribution calculations provided by two regional models (EMEP and LOTOS-EUROS) in a forecast mode. Together, the models compose the operational source contribution prediction system for the European cities within the Copernicus Atmosphere Monitoring Service (CAMS) and aim to estimate the impact of the long-range transport to urban

$PM_{10}$. These models also use two distinct source apportionment methodologies, a labeling technique for LOTOS-EUROS and the use of perturbation runs for EMEP.

The methodology used for the EMEP model was tested by using three different percentages (5%, 15% and 50%) in the perturbation runs. The importance in the choice of the domain defining the edges of the studied cities was also investigated in terms of predicted concentrations and calculated contributors. It was concluded that the 15% emission reduction and the use

of large city areas (9 grids or GADM) were the more efficient. It reduces the impact of non-linearity, which especially impacts the $NO_3^-$, $NH_4^+$ and $H_2O$ concentrations, and it presents a better agreement in the determination of main country contributors. The mean non-linearity always represents less than 2% of the total modelled $PM_{10}$ for each contribution calculated by the EMEP SC and is caused by the perturbation used, which is assumed to be linear to a 100% perturbation. Even if this non-linearity is not identical for all cities and for the different dates, the larger non-linearities (>5%) impact only 3% of all the

calculated hourly contributions. However, the non-linearity related to the reduction of each emission precursor has not been calculated in the study for computational reason.

The predicted $PM_{10}$ concentrations were compared with AirBase observations, showing fair agreement even if the models remain perfectible since they have difficulties to reproduce the highest hourly concentrations measured by the urban stations (mean underestimation by 36%). It may suggest that both models, which calculate the country contributions over the cities, defined by a large area, may underestimate the contribution measured by a specific station for the higher concentrations. It was also noticed the bias is slightly reduced when the forecast is closer to the studied date. An inter-comparison between both models was also performed showing satisfactory results with few discrepancies in the predictions of the $PM_{10}$ concentrations, mainly explained by an underestimation in sea salt and dust by the EMEP model (compared to LOTOS-EUROS); and differences in SIA, caused by different chemical aerosols treatment in both models.

During the episode, both models have shown that 45% of the predicted $PM_{10}$ over the selected cities were from "Domestic" sources and essentially composed of primary components. The rest of the contribution was roughly equitably split into an influence from the others 30 countries used in the regional domain, essentially composed of $NO_3^-$ and from "Others" mainly composed of natural sources.

We have shown that results from both source apportionment methodologies agree in average by 68% in the determination of the dominant country contributor to the hourly $PM_{10}$ concentrations and 75% for the top 5 of these country contributors. Calculating the country attribution on a daily-mean basis has similar agreement. Where there are differences, these are mainly found in the country attribution of the secondary inorganic component of the aerosol. These differences derive from a combination of the different treatment of these secondary components and the different method to attribute country contributions between the models being compared.

A full year of evaluation will be necessary to confirm our satisfactory results. Moreover, the bias of the predicted $PM_{10}$ concentrations with the urban observations probably suggests an underestimation of the "Local" background contribution (from the city) which is also predicted by the EMEP model. This is investigated in a companion paper (in preparation), also focusing on the same event.

**Data availability**

The EMEP model is an open source model available on https://doi.org/10.5281/zenodo.3355041. The base-code of LOTOS-EUROS is available under the license on https://lotos-euros.tno.nl/, but the code used for this study, including the source apportionment is only available in cooperation with TNO. The data processing and analysis scripts are available upon request.


**Author contribution**

MP, HF and M Schulz designed the research. MP performed the experiment. MP developed the analyzing codes and analyzed the data. AV developed the EMEP part of the forecasting system. RK and M Schaap performed and provided the LOTOS-EUROS results. MP wrote the paper with the inputs from all coauthors.

**Acknowledgments**

This work is partly funded by the EU Copernicus project CAMS 71 to provide policy support. This work has also received support from the Research Council of Norway (Programme for Supercomputing) through the EMEP project (NN2890K) for CPU and the Norstore project "European Monitoring and Evaluation Programme" (NS9005K) for storage. The EMEP project itself is supported by the Convention on the Long Range Transmission of Air Pollutants, under UN-ECE. The authors thank

A. Mortier (Norwegian Meteorological Institute) for the development and the design of the website (https://policy.atmosphere.copernicus.eu/SourceContribution.php).

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

**Table. 1** Technical description of both models used in the SC calculation system.

| Model | EMEP/MSC-W | LOTOS-EUROS |
|---|---|---|
| Model version | rv4.15 (open source version Sept 2017) | V2.0 (open source version 2016) |
| Horizontal resolution | $0.25° \times 0.125°$ lon-lat | $0.25° \times 0.125°$ lon-lat |
| Regional domain | 30°N-76°N<br>30°W-45°E | 31°N-68.875°N<br>24°W-43.75°E |
| PBL | Calculation based on turbulent diffusion coefficients (Kz) (EMEP Status Report 1/2003) | From ECMWF |
| Vertical resolution | 20 sigma layers up to 100 hPa, with about 10 in the Planetary Boundary Layer | Mixing layer approach with a 25m surface layer. Model top at 5 km. |
| Gas phase chemistry | Evolution of the "EMEP scheme" (Andersson-Sköld and Simpson, 1999; Simpson et al. 2012) | TNO-CBM-IV (Schaap et al., 2009) |
| Nitrate formation | Oxidation of $NO_2$ by $O_3$ on aerosols (night and winter)<br>$N_2O_5$ hydrolysis on aerosol (Simpson et al., 2012) | $N_2O_5$ hydrolysis on aerosol (Schaap et al., 2004) |
| Sulphate production | $SO_2$ oxidation by $O_3$ and $H_2O_2$ | $SO_2$ oxidation by $O_3$ and $H_2O_2$ |
| Inorganic aerosols | MARS (Binkowski and Shankar, 1995) | ISORROPIA-II (Fountoukis and Nenes, 2007) |
| Secondary organic aerosols | EmChem09soa (Bergström et al, 2012) | Not included in this model version |
| Water | $PM_{10}$ particle water at 50% relative humidity | Not diagnosed |
| Advection | Scheme of Bott (1989) | Monotonic advection scheme (Walceck and Aleksic, 1998) |
| Dry deposition/sedimentation | Resistance approach for gases and for aerosol, including non-stomatal deposition of $NH_3$ (EMEP Status Report 1/2003) | Resistance approach for gases and for aerosol, including compensation point for $NH_3$ (van Zanten et al., 2011; Wichnik Kuit et al., 2012; Zhang, 2001) |
| Wet deposition | wash out ratio's | pH dependent wash out ratio's accounting for saturation |
| Dust | Boundary conditions + windblown dust | Boundary conditions + Soil, traffic and agriculture (Schaap et al., 2009) |
| Sea Salt | Mårtensson et al. (2003), Monahan (1986)<br>production accounting for whitecap area fractions (Callaghan et al., 2008) | Mårtensson et al. (2003), Monahan (1986) |
| Boundary values | global C-IFS 00UTC | global C-IFS 00UTC, except for sea salt |
| Initial values | 24h forecast from the day before | 24h forecast from the day before |
| Anthropogenic emissions | TNO-MACC-III for 2011 | TNO-MACC-III for 2011 |
| Fire emissions | CAMS product: GFAS | CAMS product: GFAS |

| | | |
|---|---|---|
| Biogenic emissions | Emission factors as a function of temperature and solar radiation (Simpson et al., 2012) | Emission factors as a function of temperature and solar radiation (Schaap et al., 2009) |
| Meteorological driver | 12:00 UTC operational IFS forecast (yesterday's) | 12:00 UTC operational IFS forecast (yesterday's) |


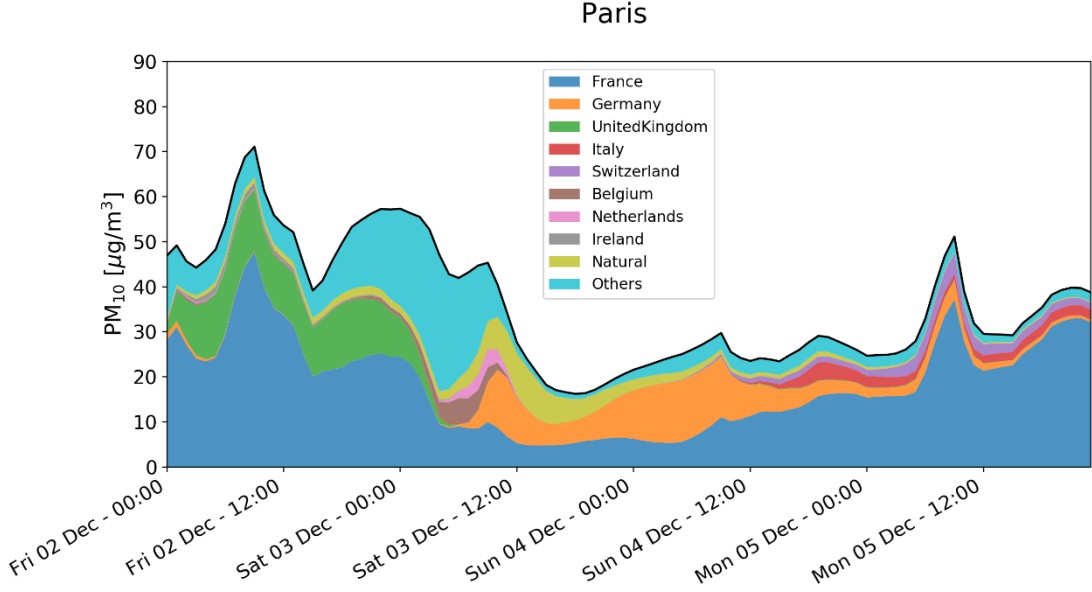

**Figure 1: Hourly PM$_{10}$ concentrations in μg/m$^3$ over Paris predicted by the EMEP model from December 2$^{nd}$ to December 5$^{th}$ 2016. The black curve highlights the total concentration. The eight main country contributors are plotted in addition to the natural sources and "Others". "Others" gathers hereafter other European countries, the boundary**
**conditions, the ship traffic, the biogenic sources, the aircraft emission and the lightning.**


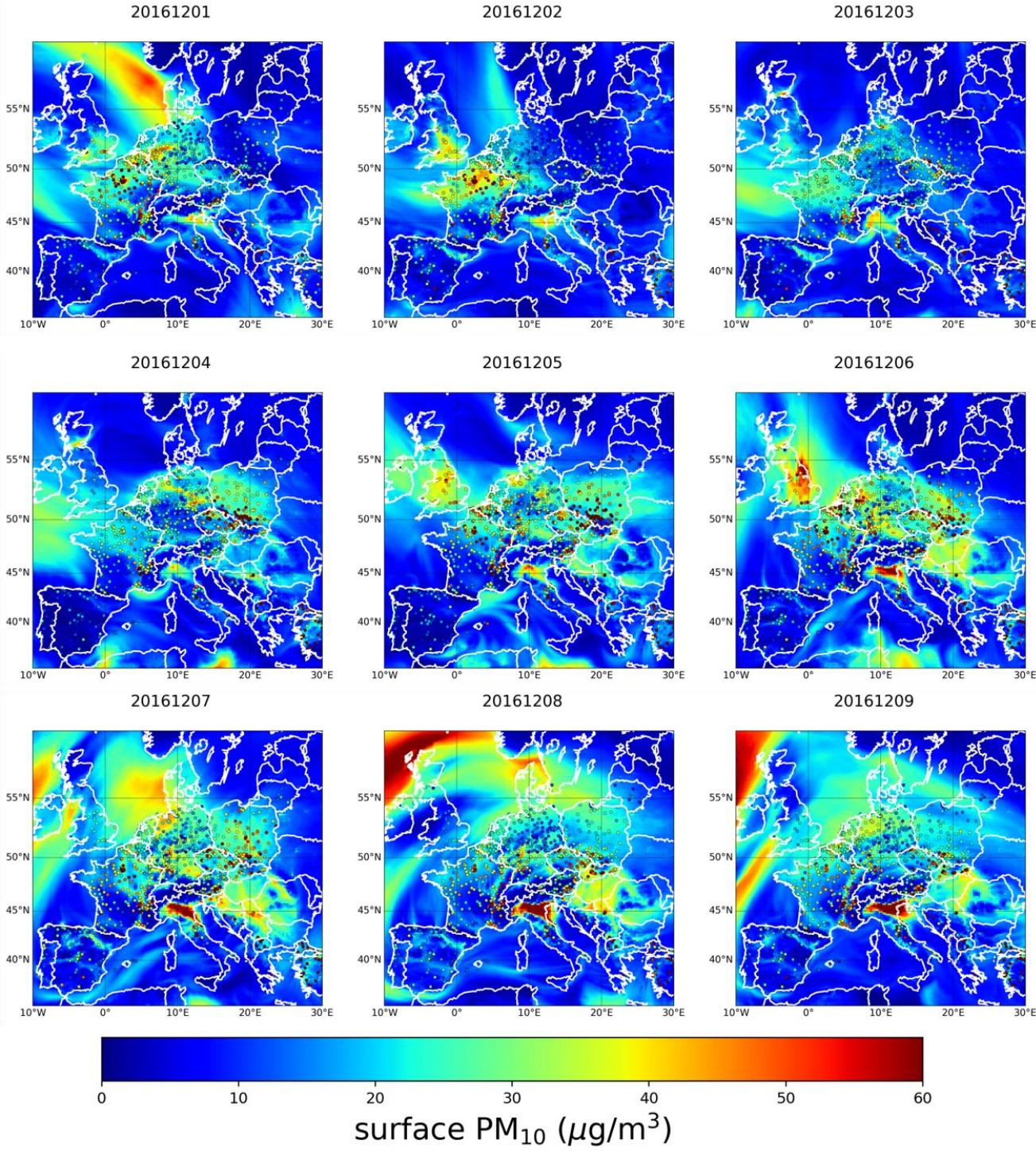

**Figure 2: Daily surface PM₁₀ concentration in µg/m³ over Europe predicted by the EMEP model from December 01ˢᵗ to 09ᵗʰ 2016. The colored dots correspond to the daily mean of AirBase stations (rural and urban stations).**

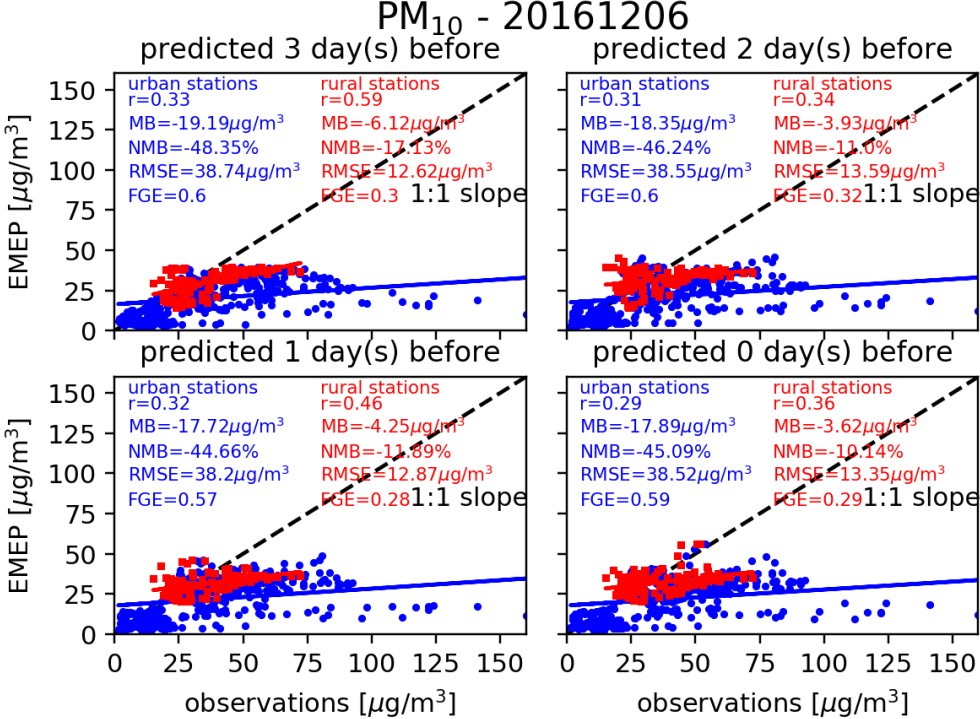

**Figure 3: Scatterplot between the hourly PM₁₀ concentrations in μg/m³ over all the studied cities using the 9 grid cells definition, predicted by the EMEP model on December 06th 2016 and the observations of the urban sites (blue dot) and rural sites (red square). For this case, there are 19 cities which have urban stations in their domain and 5 cities which have rural stations in their domain. The observations are collocated in time to the EMEP predictions and then averaged within the city edge to match the studied grid. The four panels correspond to the different predictions from 3 days before the December 06th to the actual day, i.e. December 06th. The correlation coefficient (r), the mean bias (MB), the normalized mean bias (NMB), the root-mean-square error (RMSE) and the fractional gross error (FGE) are provided on each panel. The blue and the red lines represent the linear fits.**

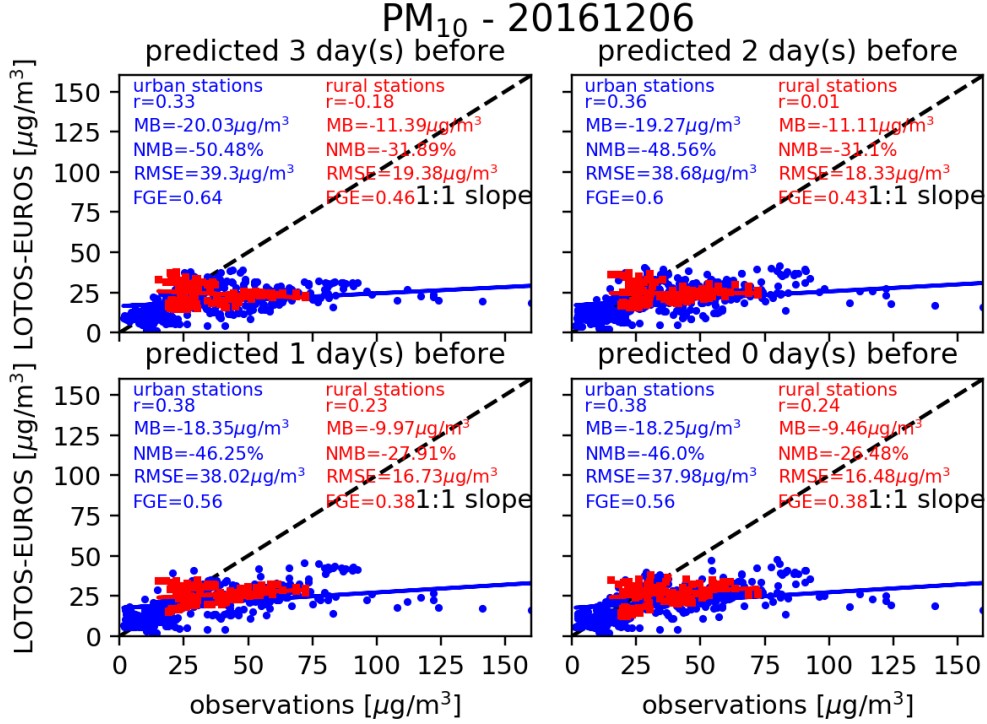

**Figure 4: As Fig. 3 for LOTOS-EUROS.**

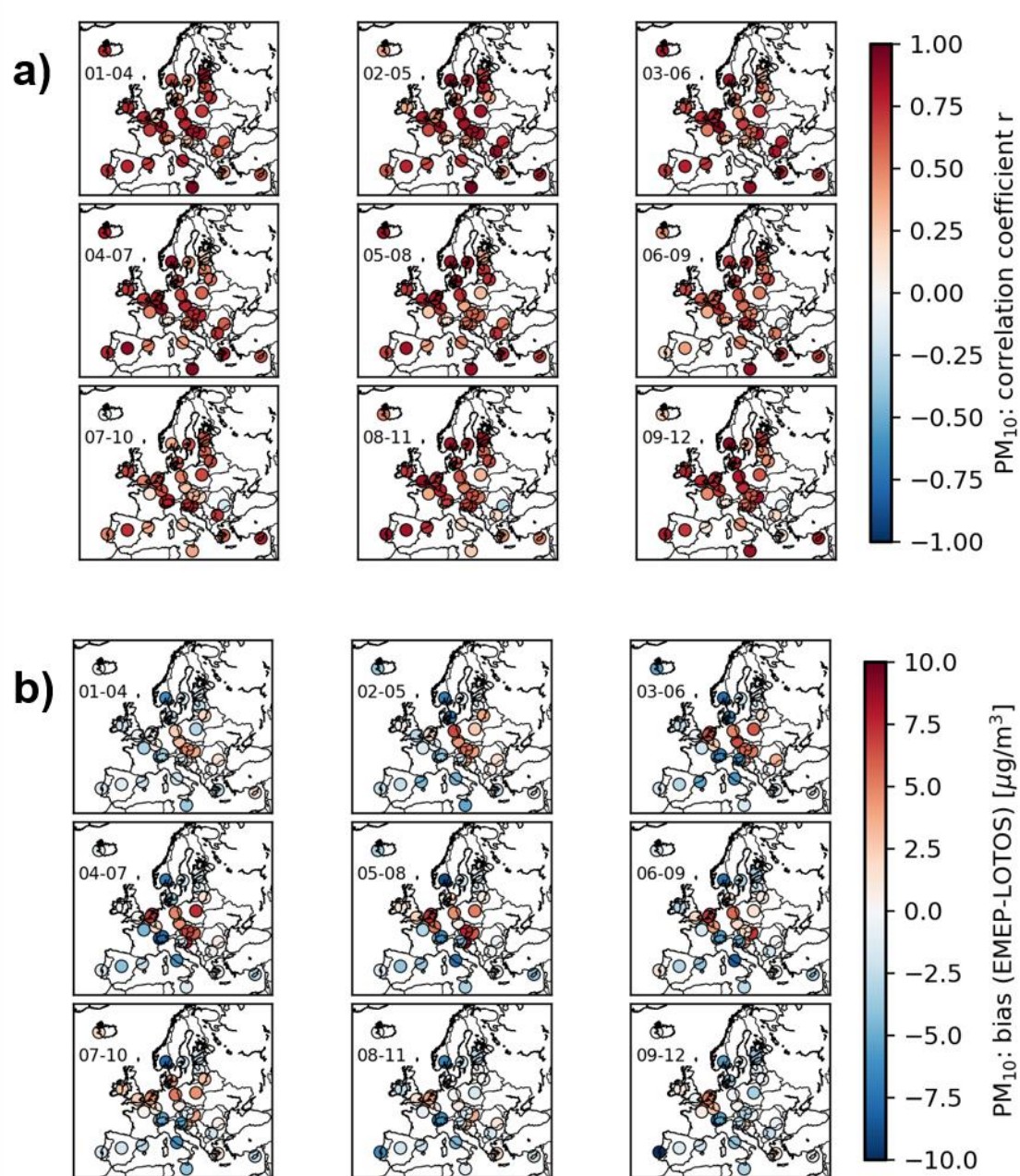

**Figure 5: Correlation coefficient (a) and bias (b) in the predicted PM₁₀ concentrations between the EMEP model and LOTOS-EUROS over all the studied cities using the 9 grid cells definition for each 4-day forecast (01-04 Dec 2016, 02-05 Dec 2016, 03-06 Dec 2016, 04-07 Dec 2016, 05-08 Dec 2016, 06-09 Dec 2016, 07-10 Dec 2016, 08-11 Dec 2016, 09-12 Dec 2016).**



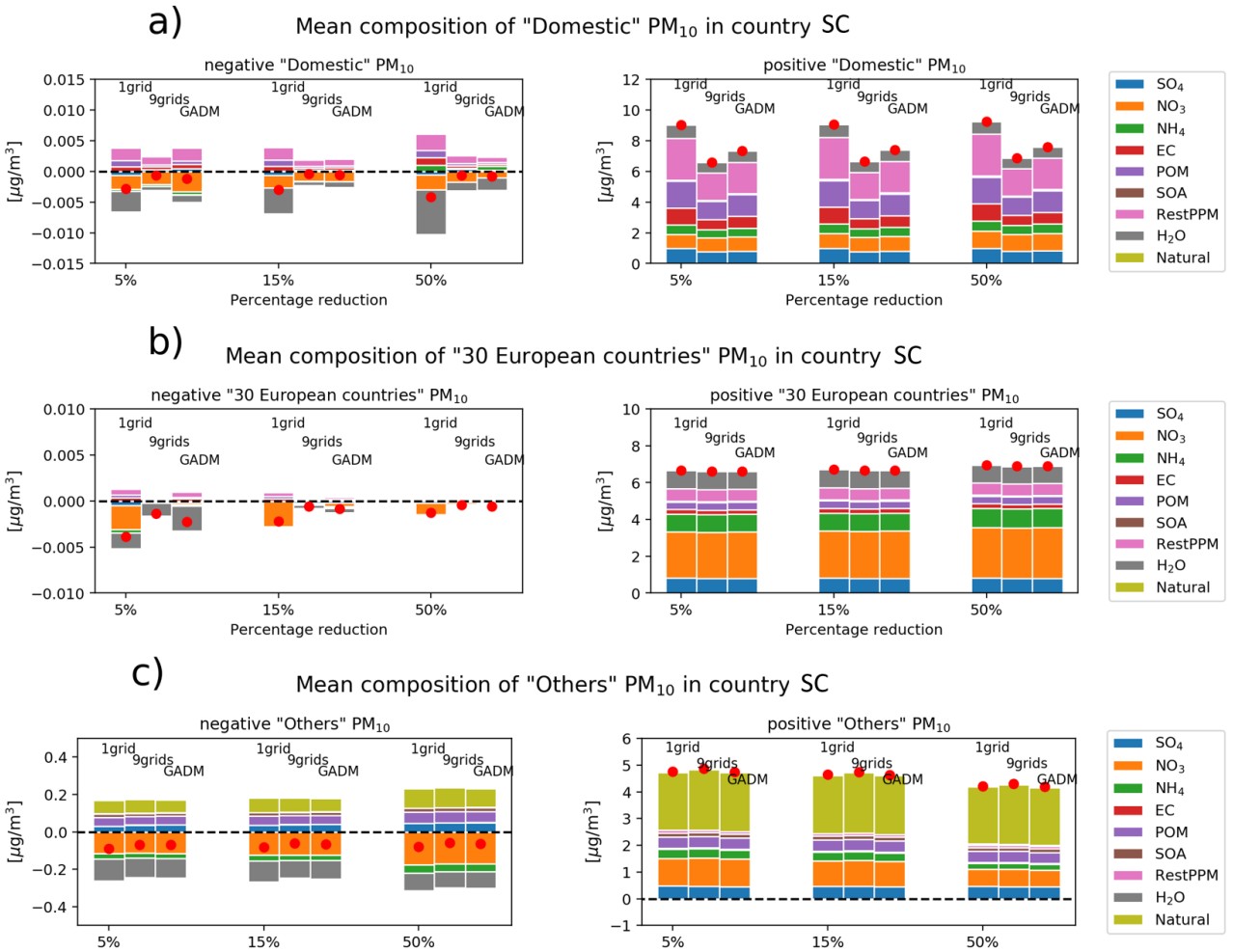

**Figure 6: Mean composition of "Domestic" (a), "30 European countries" (b), and "Others" PM$_{10}$ split into a negative concentration (left panel) and a positive concentration (right panel), calculated by the EMEP country SC over the 34 European cities and for each 4-day forecast. The PM$_{10}$ composition is highlighted with the color code. The results for the 3 city definitions (1 grid, 9 grids, GADM) and for the percentage of reduction used in the perturbation EMEP runs (5%, 15%, 50%) are shown. The "Domestic" contribution corresponds to the contribution from the domestic country to the city (e.g. from France to Paris). "30 European countries" corresponds to the other 30 European countries used in the study. "Others" gathers natural sources, the other countries included in the regional domain, the boundary conditions, the ship traffic, the biogenic sources, the aircraft emission and the lightning. The red dot represents the mean PM$_{10}$ concentration.**

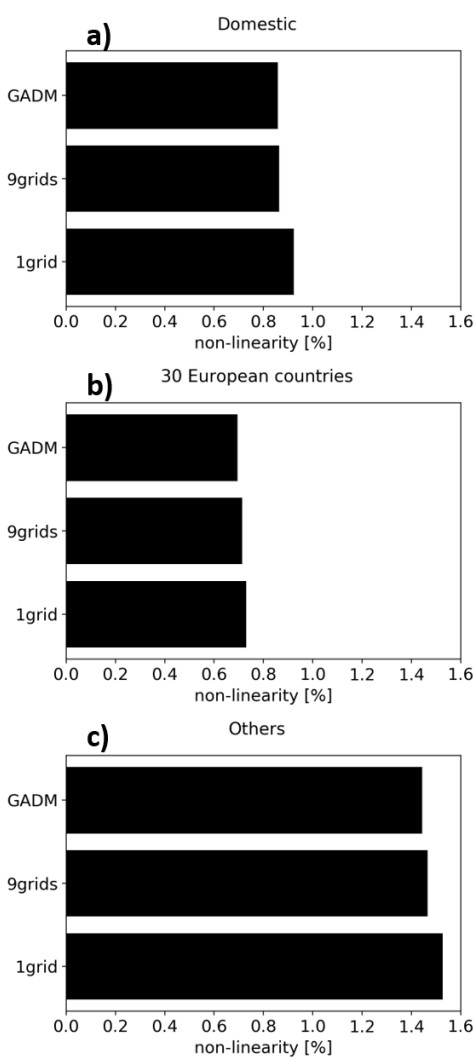

**Figure 7: The black horizontal bars show the mean non-linearity calculated for each contribution presented in Figure 6 and for the three city definitions. The non-linearity is calculated for each hourly concentration as the standard deviation of the hourly contribution weighted by the hourly total concentration.**

a)

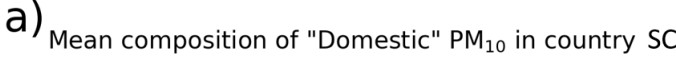

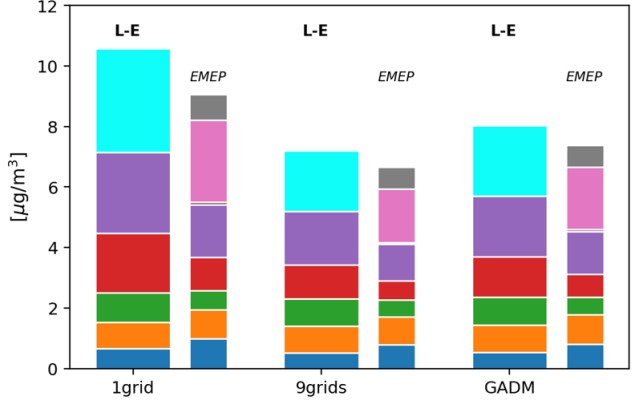

b)

Mean composition of "30 European countries" PM$_{10}$ in country SC

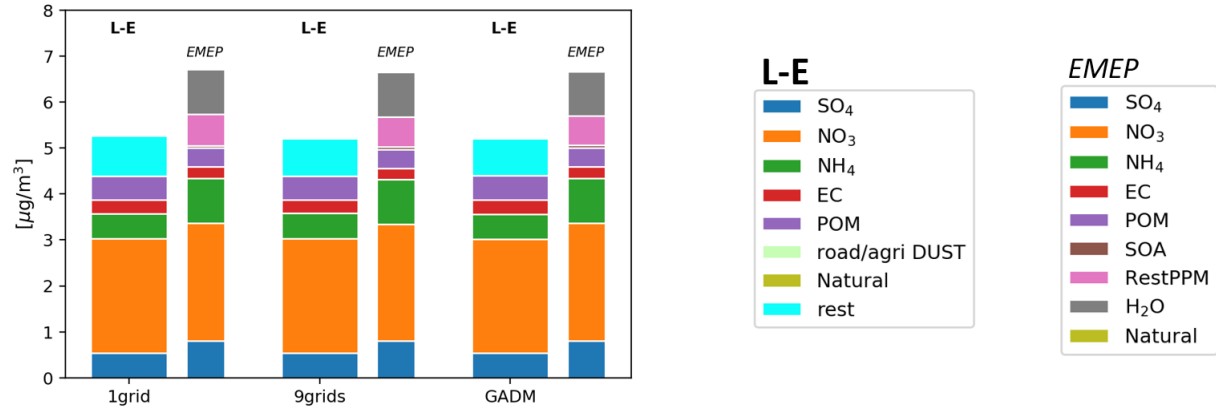

c)

Mean composition of "Others" PM$_{10}$ in country SC

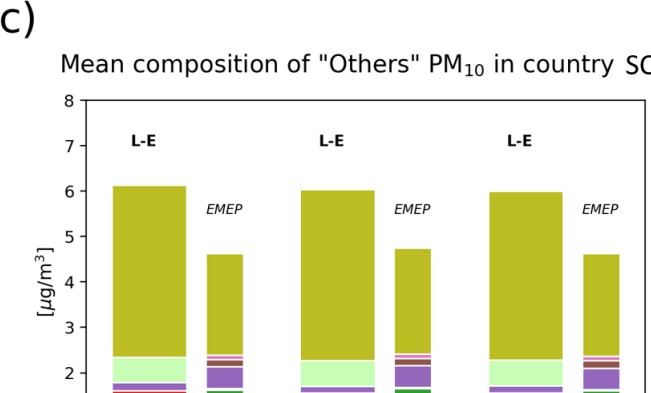

**Figure 8: Mean composition of "Domestic" (a), "30 European countries" (b), and "Others" PM$_{10}$ calculated by the LOTOS-EUROS (L-E) country SC over the 34 European cities and for each 4-day forecast. The result from the EMEP country SC, by using a 15% perturbation run has also been added for comparison. The PM$_{10}$ composition is highlighted with the color code. Rest corresponds to the difference between the PM$_{10}$ and the sum of the components listed on the plot. The results for the 3 city definitions (1 grid, 9 grids, GADM) are shown. The "Domestic" contribution corresponds to the contribution from the domestic country to the city (e.g. from France to Paris). "30 European countries" corresponds to the other 30 European countries used in the study. "Others" in the LOTOS-EUROS country SC is slightly different to the EMEP "Others". "Others" in the LOTOS-EUROS country SC gathers natural sources, the other countries included in the regional domain, the boundary conditions, the dust emitted by the road traffic and agriculture, the ship traffic, the aircraft emission and the lightning.**

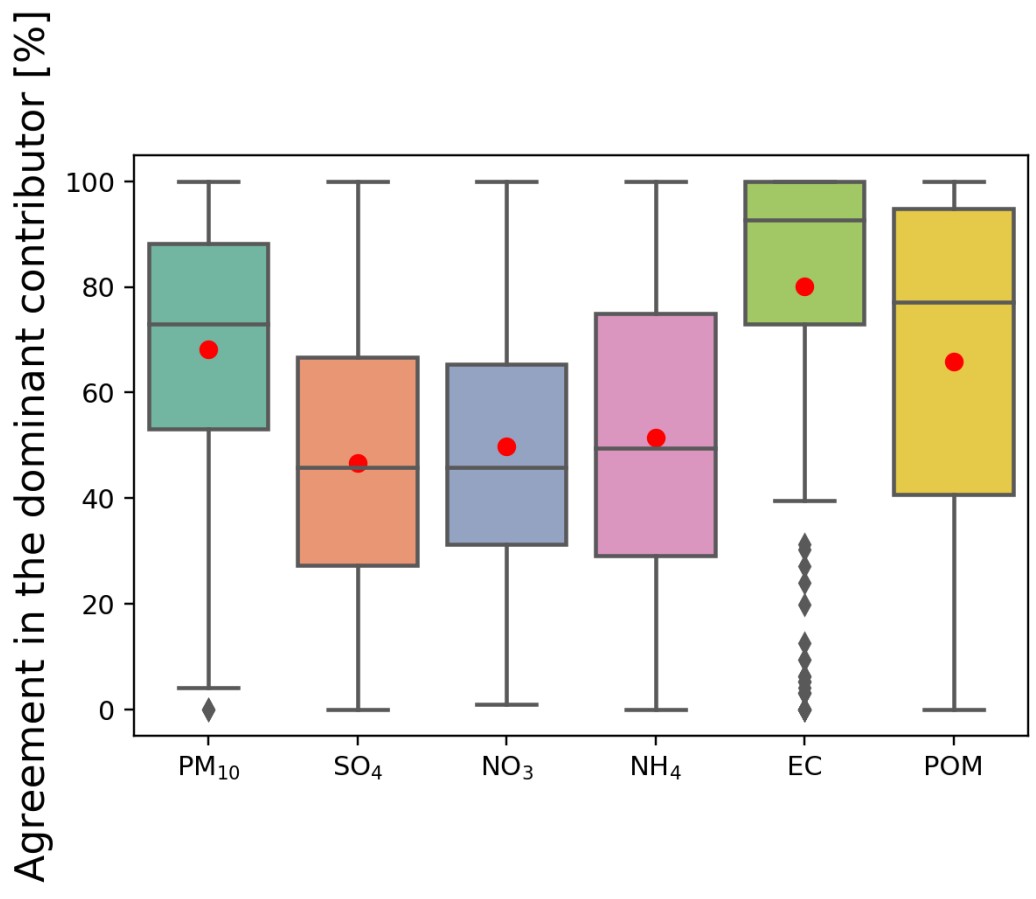

**Figure 9: Agreement in the determination of the dominant country contributor for PM$_{10}$, SO$_4$, NO$_3$, NH$_4$, EC and POM in percent, determined over all the studied cities using the 9 grid cells definition and for all 4-day forecasts. The line that divides the box into two parts represents the median of the data. The end of the box shows the upper and lower quartiles. The extreme lines show the highest and lowest value excluding outliers which are represented by grey diamonds. The red dots correspond to the mean of each data set.**



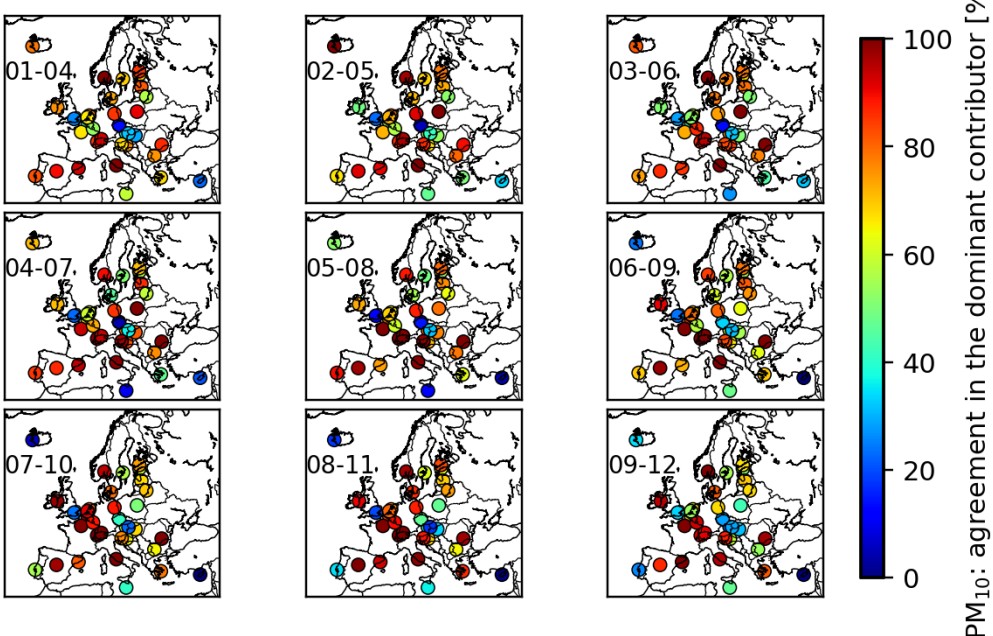

**Figure 10: Agreement in the determination of the dominant country contributor for PM₁₀ in percent, and for each 4-day forecast (01-04 Dec 2016, 02-05 Dec 2016, 03-06 Dec 2016, 04-07 Dec 2016, 05-08 Dec 2016, 06-09 Dec 2016, 07-10 Dec 2016, 08-11 Dec 2016, 09-12 Dec 2016) over all the cities using the 9 grid cells definition.**


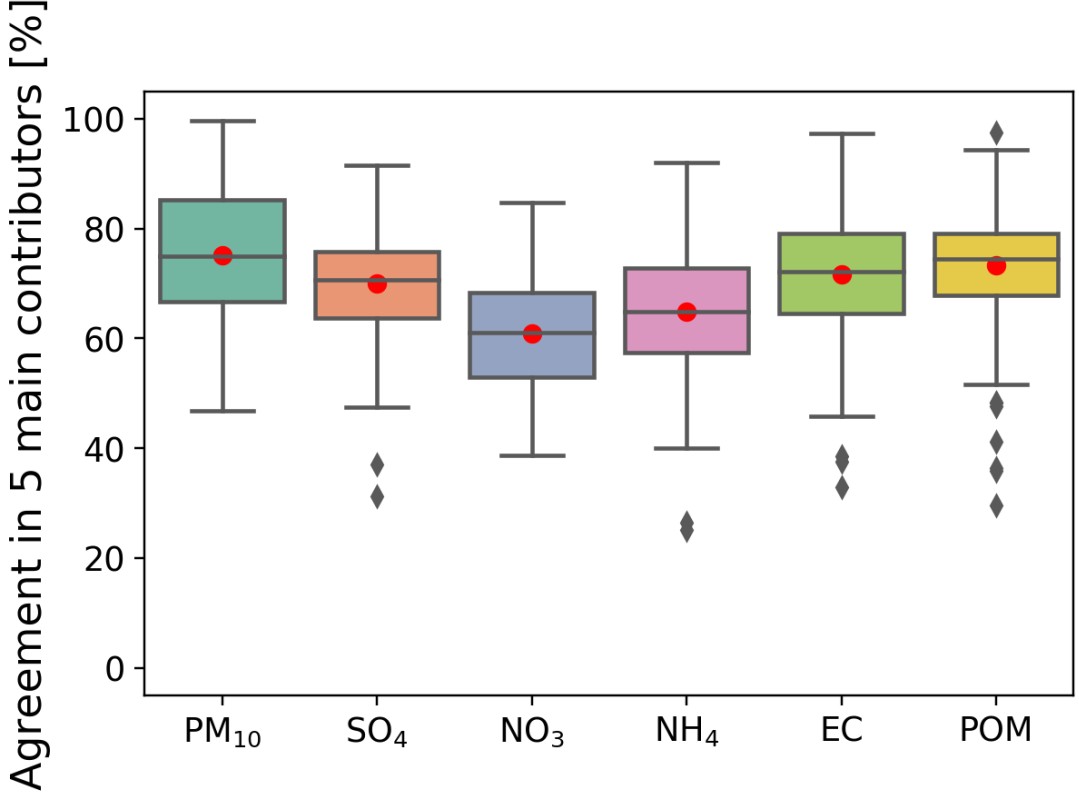

**Figure 11: Agreement in the determination of the five main country contributors for PM₁₀, SO₄, NO₃, NH₄, EC and POM in percent, determined over all the studied cities using the 9 grid cells definition and for all 4-day forecasts. The line that divides the box into two parts represents the median of the data. The end of the box shows the upper and lower quartiles. The extreme lines show the highest and lowest value excluding outliers which are represented by grey diamonds. The red dots correspond to the mean of each data set.**
