# Peer review of "Prediction of source contributions to urban background PM10 concentrations in European cities: a case study for an episode in December 2016 by using EMEP/MSC-W rv4.15 and LOTOS-EUROS v2.0 - Part.1 The country contributions"

_Geoscientific Model Development, 2019_

## Referee Comment (RC1) · Anonymous Referee #1 · 16 Jun 2019

The study by Pommier et al., demonstrates the ability of two modelling setups to identify source contributions of particulate matter from different countries to multiple capital cities in Europe during a pollution episode. Overall, the paper does indeed demonstrate this and after some major revisions it should be suitable for publication in GMD.

The main concerns I have with the manuscript is its lack of clarity in places. Firstly, the description of the source-receptor calculations needs to be more clearly discussed as it is not easy to follow to non-experts of this methodology. Secondly, the manuscript is compiled of lots of short sentences which lead to a stop-start flow with makes the

manuscript more difficult to read. Thirdly, the comparisons between the models and the observed PM concentrations are satisfactory at best. For instance, many of the correlation statistics between the models and observations are below 0.5 and the mean biases (magnitude) are sizable. Therefore, I feel the authors really need to stress that the model comparisons are "satisfactory" and clearly state whether the metrics presented (e.g. P11) show the models are doing well or badly when compared to the observations. Finally, some of the figures are too busy and need to be made clearer. For instance, Figure 6 is overly complicated and takes a long time to fully digest. Also, the "agreement in the dominant contributor" in figure 8 between the models is not clear. How is this agreement determined? What statistical metrics are used? If this is already stated, then please make it clearer!

Minor Comments: P3 L71-73: Provide reference for the WHO health metric stated. P3 L84: Space between "VOCs).The". P4 L99-101: Please explain in detail how "source" and "receptor" are related in this work to make it clear for readers not familiar with this method. P5 L158: What is the new land-cover dataset used? P6 L168-169: Make it clear whether or not other BVOC emissions are used in the model other than iso-prene and monoterpenes. P6 L170: The definition of the "remainder" is unclear. P7 L212-213: What do the authors mean by "we have harmonized the used of different parameters"? Do you mean that the setup and input/outputs of the model are been made as consistent as possible? P8 L 222: Worth saying that the ECMWF operational system does not archive 3D precipitation fields when this is first discussed on P6. P8 L 247: Can the authors please elaborate on what they mean by "medium intensity". P9 L256: What quantifies as "large concentrations"? P10 L277: I suggest the authors change the word "enormous" to something more scientific. P11: L307-312: I suggest the authors re-write this paragraph as it is unclear and difficult to follow. P11: The discussion of the different metrics is a bit over-kill here. If all this discussion is to be kept in the manuscript, can the authors at least specify what the numbers mean in terms of model performance (e.g. R=0.72 is reasonable and R=0.25 is poor). P13 L365-371: As mentioned above, I think the authors need to discuss in more detail the

source-receptor methodology to make it clear to non-experts of this approach. P14 L415: "For the positive correlation, a clear feature appears" is an example of these short sentences which break the flow of the text. P15 Section 5.2: If the LOTUS model is using a different approach to that of EMEP, how are the emissions perturbed? This is not overly clear from the text as it stands. P16 L458: What do the authors mean by "each at the end of the EMEP model"? P16 L459-462: The term "Rest" appears to represent the difference between the total PM and the sum of all its components. Is this the metric used to explain the "non-linearity in the chemistry? If so or if not, I think this sentence need to be rewritten to clear emphasis the definition of "Rest". P16 L467-8: Is this true? In section 3 I got the impression there was substantial disagreement between the models. Figure 2: Could the country outlines be more clearly plotted. Figure 6: There is a lot of stuff is this plot, so could be good to make it simpler or bigger at least so easier to see everything. The calculation of non-linearity need to be explained more clearly in the manuscript.

―――――――――――――――――――――

---

## Short Comment (SC1) · 8 Jul 2019

In this publication, the Authors compare two different source apportionment approaches (referred as "scenario" and "labeling") and conclude that they reach a satisfactory agreement (68% for PM10, 50% for SIA) between the two methods for a few-days episode analyzed in more than 30 cities. This claim of a satisfactory agreement is based on numbers that represent average results across cities and forecast days. While this average agreement/non-agreement probably represents a necessary first step, it is not sufficient in my view. Given the capacity of air quality models to deliver highly resolved data in terms of space and time, a user will mostly be interested in the results for a specific city and specific day. When looking at results detailed in terms of city and forecast day (Figure 9 and box-plots 8 and 10), the agreement is quite low for some cities/days. Moreover, a low agreement between the two methods for some cities/days is not surprising. Indeed, conceptual differences do exist between the scenario and tagging/labeling approaches that have been shown to generate important differences in terms of results for non-linear compounds (see for example Burr and Zang 2011, Grewe et al. 2010, Thunis et al. 2019). This point was also made by Kranenburg et al. (2013) themselves. In their work, Clappier et al. (2017) and Grewe et al. (2010) explained conceptually why these two approaches do not lead to comparable results for non-linear compounds and concluded that these two methods should serve different purposes. Given the above points, I find the Author's conclusions surprising and also misleading in terms of their implications on air quality management practices as they suggest that both methods are equally suited for calculating source contributions (see e.g. lines 113-114) when this is known not to be the case.

A few other points are raised below.

- As shown by Clappier et al. (2017) or Kranenburg et al. (2013), the results of the scenario and labeling techniques would lead to identical results for the linear fraction of PM (primary), if obtained with the same underlying air quality model. The level of agreement obtained for primary compounds like EC therefore provides quantitative information on the difference caused by the underlying model (LOTOS vs. EMEP). On the other hand the difference in agreement between primary and secondary (NO3, SO4 or NH4 Figures 8 and 10) is a direct consequence of the apportionment of the secondary fraction which conceptually differs in the two methods. The lower agreement for SIA than for primary is not only due to differences between EMEP and LOTOS as suggested at lines 477-478, but also, according to me, because of the conceptual differences between the labeling and scenario approaches.

- The impact of the reduction percentage used in the scenarios is shown as an average

over cities and forecast days (Figure 6). It is unclear how the average indicator has been obtained (have negative and positive differences been summed-up in absolute term?) but even if in absolute terms, the average process does not show the real level of non-linearity obtained for specific cities and days. Thunis et al. (2016) have shown, based on LOTOS_EUROS simulations, that non-linearities could reach more that 5 to 10% on daily values and that the "interaction non-linearity (ignored in the current work) was the dominating factor (up to 20% in some cities). The level of non-linearity obtained here (around 1%) is very low and therefore surprising. It would interesting to see detailed values of this non-linear indicator for each city/day.

- When noting that the results of scenario and labeling differed for non-linear species, Kranenburg et al. (2013) compared a 5% scenario reduction with a simulation where only 5% of the emissions where labeled. Could the Authors explain why they did not label only 15% of the emissions in this comparison?

References:

Grewe V., E. Tsati and P. Hoor, 2010. On the attribution of contributions of atmospheric trace gases to emissions in atmospheric model applications, Geosci. Model Dev., 3, 487–499.

Burr M.J. and Y. Zhang, 2011. Source apportionment of fine particulate matter over the Eastern U.S. Part II: source sensitivity simulations using CAMX/PSAT and comparisons with CMAQ source sensitivity simulations, Atmospheric Pollution Research, 2, 318-336.

Thunis P., A. Clappier, E.Pisoni, B.Degraeuwe, 2015: Quantification of non-linearities as a function of time averaging in regional air quality modeling applications, Atmospheric Environment, 103, 263-275.

Clappier A., C. Belis, D. Pernigotti and P. Thunis, 2017: Source apportionment and sensitivity analysis: two methodologies with two different purposes. Geosci. Model

Dev., 10, 4245-4256.

Kranenburg R., Segers A., Hendriks C., and Schaap, 2013. Source apportionment using LOTOS-EUROS: module description and evaluation, Geosci. Model Dev., 6, 721–733.

---

## Referee Comment (RC2) · Anonymous Referee #2 · 9 Jul 2019

This paper compares two source apportionment methods. The methods are not clearly explained. Some clarifications are needed and there are some methodological flaws. Also the English used in this paper needs to be revised.

Specific comments: 1. About the comparison between measured and modeled concentration. I understand that the author wants the compare the average concentration over an urban area. From model results it is easy to obtain this, averaging concentrations over some grid cells. Unfortunately you cannot obtain a comparable number from measurements. The stations are not equally distributed over the area of interest

and the number of urban, rural and traffic stations might be different. Comparing an average of the stations with an average of the grid cells will introduce an additional uncertainty. Why not interpolate the model results at the station locations and compare with the measurements. A separate comparison for different station types should be made. I think the analysis for one cell, 9 cells and the GADM. I would restrict the analysis to stations inside the GADM.

2. The non-linearity discussed Line 374 and following The contributions of individual countries don't have to theoretically sum up to the contribution of all countries reduced together. Even for small reductions there is some non-linearity. But the non-linearity is small for small reductions. The difference between the sum of individual contributions and the joint contribution can be positive or negative. I would not speak about negative concentrations. You scale up to 100% but in fact you do a source apportionment of the top 15% of the PM10 column. That's perfect and useful for policy. Achieving small emission reductions is already hard enough.

3. Validation versus measurements The validation shows quite big differences between model and measurements. What is the impact of this error on the source apportionment? To which extent can it be trusted? In regard of this error, which differences between the two methodologies are significant? How certain is it that the biggest contribution is really the biggest?

4. Figure 6 Maybe it is more useful to present the analysis for some selected cities (and the others in Annex) than for all cities together. The behavior can be quite different across Europe. If non-linearity is small plots for one reduction percentage are sufficient. It is not clear to me which runs were done to obtain these plots. On line 358 is mentioned that emission per country where reduced with 15%. Are precursors reduced one by one or all together? How is the non-linearity calculated? Is it calculated as a share of the total concentration (Line 506). In my opinion it is more correct to use the concentration change as reference?

5. Comparability of the two methodologies. For primary pollutants both source apportionment methodologies are comparable. Differences are due to differences in the models (transport, deposition,..). But for secondary PM the methods don't necessarily give the same result. E.g. an amount of NOx emitted somewhere can result in a certain ammonium nitrate concentration in the receptor. If NOx is emitted in excess (ammonia limited regime) an emission reduction will have little effect at the receptor point. On the other hand, in the NOx limited regime the same NOx reduction will have a big impact. The labeling method will give the same result in both cases while the 'perturbations' method will give different results. Hence, comparing contributions calculated by the two models is not very useful. The statement on line 513 is not complete: differences are not only due to differences in aerosol chemistry between the models.

6. Figure 8 How is the percentage of agreement defined? I think it's more useful to present this for individual cities.

Technical comments: Line 30: change '15% factor' > '15% emission reduction'. This 15% is not a factor. I think it's confusing. Change this in the whole paper. Line 30-33: revise grammar, sentence too long. We found that the combination of a 15% reduction and a larger domain help to reduce... Line 36: split sentence Line 68: crops yields > crop yields Line 71: states > better established/proposed a PM10 limit value Line 77-79: very unclear contradictory sentence. If a pollutant has a short life time it's impact is close to the source and long-range transport doesn't matter much. Is PM10 really so short lived compared to other pollutants (like NO2). The concentration of PM10 is rather uniform compared to the latter. Line 81: atmospheric processing? > formed by chemical reactions in the atmosphere. Line 85: traffic and transport, all traffic is transport Line 86: biomass burning refers to burning wood for heating. It is an anthropogenic source. You mean wild fires? Line 95: Revise grammar and content. With a country source calculation...???? Line 98: revise. Something like: The EMEP calculations use reductions of anthropogenic emissions... Line 115: Both models are part of... Line 122: Use a consistent terminology. You say 'SR system'

and the next paragraph is called 'SA system'. A source receptor (SR) model is not a source apportionment (SA) system. Check this through the whole paper. Line 123 and 126: SR? I think it you mean source apportionment. Line 130: be consistent. SR product (should be source apportionment product). . . real-time source allocation (=? source apportionment) Line 132: for the 28 EU capitals, plus Bern, Oslo and R. Line 145: too long, split up Line 153: . . .but the model has also been used. . . Line 161: sigma coordinates? There are. . . > grammatically incorrect sentence Line 212: word order! . . .cover a slightly different domain. . . Line 119: . . .by the IFS Line 236: 1 (grid) cell . . . 9 (grid) cell. . . There is only one grid. Line 237: The latter. . . Line 238: . . . living area. . . better 'urban area' or 'build up area' Line 244: BCs ? boundary conditions? Line 250: repetition Line 255: until fronts moved in Line 258: metrics Line 259: To properly estimate Line 267: N is the number of the reference dataset? The number of what? Hours? Days? Line 283: grid cell Line 285: city edge > city boundary Line 298: . . .smoothed over a large domain. . . Do you mean smoothed over a grid cell? Mis-interpretation > underestimation. So, the correlation is similar for urban and rural stations but urban stations have a bigger bias. That's because peaks are smoothed out over the full cell. Line 303: By comparing only the 5. . . remove the comma Line 309: grammatically incorrect Line 312: . . .than the ones from the EMEP model Line 315: globally > In general Line 318: . . .at the urban. . . Line 332: negative correlation coefficients? Can you explain this better? Line 393: only . . . as well as. . . confusing formulation Line 401: . . .one source area. Line 445: Averaging out over more cells reduces non-linearity. I would not use the term 'negative concentrations' Line 454: confirms the global feature > shows the same trend (?) Line 475: reformulate Line 522: . . .probably foresees an underestimation. . . unclear formulation.

---

## Short Comment (SC2) · 26 Jul 2019

In this article, the authors compare two different source apportionment methods, both able to evaluate how different emission sources contribute to the formation of PM concentrations. The first method is a scenario approach method. It is implemented using the EMEP model to calculate the impact of the reduction of each individual source. The second method is a labelling approach. It is implemented using the LOTOS-EUROS model to calculate the contribution of different sources tracing the mass of the emitted pollutants throughout the different processes computed by the model. The authors

explain that the two methods are comparable only if the concentrations changes related to the scenario approach are not impacted by the non-linearity: Lines 111 to 114 "This highlights the importance to estimate the reliability of both methodologies in the attribution of sources to PM10 concentrations, e.g. to ensure that the concentrations changes related to the scenario approach are not impacted by the non-linearity and to show that both methodologies present similar results." I have a first serious concern with the way the authors are testing the linearity using the scenario approach: In their article Thunis P. and Clappier A. (2014) show that the non-linearity between emissions and concentrations can affect the impact of the reduction of each individual emission precursors (the concentration reduction is not proportional to the emission reduction) as well as the impact of the reduction of all the emission precursors (the concentration reduction resulting from the reduction of all the precursors simultaneously is not equal to the sum of the concentration reductions resulting from each individual precursor emission). To test the linearity the authors performed different simulations with EMEP reducing of 5, 15 and 50% all the precursors simultaneously. They claim that reducing the emissions simultaneously or separately may lead to a slight different results. Lines 383 to 384: "Furthermore, by reducing the emissions simultaneously or separately may lead to a slight different result in the concentrations, but as mentioned previously, this effect is not addressed in this work for computational reason." How can they claim that the difference between simultaneous reductions and individual reductions is slight. They did not show any results of such test which quantify this difference. Thunis P. et al (2015) show that the non-linearity resulting from the interactions between the different emission precursors is higher than the non-linearity resulting from different reduction percentages. The test performed by the authors can evaluate only a part of the non-linearity which is most likely not the most important part. This test is clearly not sufficient to evaluate the degree of non-linearity. If I refer to what the authors claim lines 111 to 114, they are unable to ensure that the scenario approach and the labelling approaches will give similar results.

I have a second serious concern with the way the authors have interpreted the conclusions of the article of Clappier et al. (2017): In their article, Clappier et al. (2017) illustrate with simple examples that the scenario approach and the labelling approaches gives similar results only if the concentrations changes related to the scenario approach are not impacted by the non-linearity for any kind of percentage reductions from 0 to 100%. This happens for non-reactive species. Clappier et al. (2017) illustrate also that, even if the scenario approach often shows linearity between emissions and concentrations for a limited reduction fraction (below 50% for example), the results provided by the scenario approach and the labelling approaches are different. That means it is expected that the two methods tested in this article will give different results, even before to start complex simulations. If I refer again to what the authors claim lines 111 to 114, they should not compare the results of the scenario approach and the labelling approaches because we know they are different. Moreover, comparing different methods using different models ensure with a great certainty that the results will be different. Then, how can we interpret the authors' conclusions? Lines 518 to 519 "It was shown that the results from both source apportionment methodologies agree in average by 68% in the determination the dominant country contributor to the hourly PM10 concentrations and 75% for the top 5 of these country contributors". Are the disagreements shown by the results due to the discrepancy between the methods or to the difference between the models?

I have a third serious concern with the way the authors interpret the capacity of the labelling and the scenario approaches to represent the reality: Lines 386 to 388 the authors mention that: "In their study, Kranenburg et al. (2013) have shown that this technique [the labelling approach] provides more accurate information about the source contributions than using a brute force approach with scenario runs as the chemical regime remains unchanged." The relation between emissions and concentrations is non linear is the real world as well as in the numerical models. If the results of the scenario approach are changing according to the percentage of reduction and/or the number of reduced emission sources, it is simply because this method is able to reflect reality. Since the reality is non-linear, the scenario approach method behaves nonlinearly. If it is used correctly the method can even quantify the degree of non-linearity. The labelling approach gives always one unique result, regardless of the degree of non-linearity of the system under study. Because they are not impacted by the non-linearity, the results are certainly much easier to show. But they give no information about non-linearity showing that the method does not reflect how the system change when the emission change. I fully agree with the authors when they write about the labelling approach: lines 108 to 109, "However, it is not designed to study the impact of emission abatement policies to pollutants concentrations...". It appears clearly that it is nonsense to claim that the labelling approach provides more accurate information about the source contributions than using a brute force approach with scenario runs as the chemical regime remains unchanged.

To conclude: This article shows significant gaps in the design of the different test as well as in the analysis of the results. I do not understand the usefulness to compare results if it is known in advance that they will be different and if it is know it will be not possible to find the origin of the differences.

Thunis, P. and A. Clappier, 2014. Indicators to support the dynamic evaluation of air quality models, Atmos. Environ., 98, 402-409 Thunis P., A. Clappier, E.Pisoni, B.Degraeuwe, 2015: Quantification of non-linearities as a function of time averaging in regional air quality modeling applications, Atmospheric Environment, 103, 263-275. Clappier A., C. Belis, D. Pernigotti and P. Thunis, 2017: Source apportionment and sensitivity analysis: two methodologies with two different purposes. Geosci. Model Dev., 10, 4245-4256.

Please also note the supplement to this comment:
https://www.geosci-model-dev-discuss.net/gmd-2019-87/gmd-2019-87-SC2-supplement.pdf
* * *
[Figure]

2019.

---

## Short Comment (SC3) · 26 Jul 2019

I am writing as executive editor to bring to your attention several ways in which this manuscript currently does not comply with GMD policy. These issues will need to be addressed before any revised manuscript could be accepted for publication.

[Figure]

Title

As per https://www.geoscientific-model-development.net/about/manuscript_types. html#item4, the names and version numbers of the models being evaluated need to be included in the title of the manuscript. Please change the title accordingly.

EMEP code on GitHub

GitHub is an excellent development platform, but it is not an archival location suitable for the code used in a paper. Indeed, even GitHub themselves tell you to use Zenodo (https://guides.github.com/activities/citable-code/). Please therefore archive (probably using Zenodo) the precise version of EMEP used in this manuscript, and cite it from the code and data availability section. The Zenodo archive will give you the entry to paste into BibTeX or another reference manager.

LOTOS-EUROS code archive

LOTUS-EUROS causes more issues because the code is not open source. It claims to be open source, however reading its licence indicates that this is not actually true because redistribution is prohibited (see https://opensource.org/osd item 1). This means that properly archiving the version of LOTUS-EUROS used will not be possible. Instead, GMD policy requires you to point out that LOTUS-EUROS is only available under a restricted licence. It remains critical that the manuscript identifies the exact version used in order to enable the results to be reproduced. The manuscript should also not make the incorrect claim that the software is open source. It is unfortunate that this claim is made on the website, however that is not a reason to reproduce the error in the manuscript.

Data availability is missing

This manuscript describes a model evaluation campaign. The models were driven using data, and evaluated using data. The code and data availability section needs to point the user at the persistent public archives for the precisely identified code that was used. For a model evaluation paper, this is likely to make this section quite expansive, in contrast with its current brief extent.

Configuration files, run scripts and evaluation scripts

Reproducibility also demands that the exact configuration files and scripts used to run the models are presented, along the scripts used to process and evaluate the model output. Please also archive and cite this data.

For a fuller description of GMD's code and data evaluation policy, please see: https://opensource.org/osd

---

## Author Comment (AC1) · 22 Sep 2019

We would like to thank Philippe Thunis for his helpful comments. We have tried to answer all his remarks below.

We also apologize for our missing points in our analysis and we are very grateful for all the details provided by Philippe Thunis.

We also would like to remind that we aimed to provide in this study, the origin of the PM under current conditions and thus to provide the information which sectors/regions to target. Effectivity of measures as such is not aimed for here.

In this publication, the Authors compare two different source apportionment approaches (referred as "scenario" and "labeling") and conclude that they reach a satisfactory agreement (68% for PM10, 50% for SIA) between the two methods for a few-days episode analyzed in more than 30 cities. This claim of a satisfactory agreement is based on numbers that represent average results across cities and forecast days. While this average agreement/non-agreement probably represents a necessary first step, it is not sufficient in my view. Given the capacity of air quality models to deliver highly resolved data in terms of space and time, a user will mostly be interested in the results for a specific city and specific day. When looking at results detailed in terms of city and forecast day (Figure 9 and box-plots 8 and 10), the agreement is quite low for some cities/days.

In Fig. 8 we have decided to show the mean agreement since to present 34 figures (one for each city) will be unreadable. However, in the preparation of our study, the agreement was also calculated for each component, as shown below:

---

## Author Comment (AC2) · 22 Sep 2019

The study by Pommier et al., demonstrates the ability of two modelling setups to identify source contributions of particulate matter from different countries to multiple capital cities in Europe during a pollution episode. Overall, the paper does indeed demonstrate this and after some major revisions it should be suitable for publication in GMD. The main concerns I have with the manuscript is its lack of clarity in places. Firstly, the description of the source-receptor calculations needs to be more clearly discussed as it is not easy to follow to non-experts of this methodology. Secondly, the manuscript is compiled of lots of short sentences which lead to a stop-start flow with makes the manuscript more difficult to read. Thirdly, the comparisons between the models and the observed PM concentrations are satisfactory at best. For instance, many of the correlation statistics between the models and observations are below 0.5 and the mean biases (magnitude) are sizable. Therefore, I feel the authors really need to stress that the model comparisons are "satisfactory" and clearly state whether the metrics presented (e.g. P11) show the models are doing well or badly when compared to the observations.

The authors would like to thank the reviewer 1 for his comments which help to improve our study. We have tried to clarify the points raised by the reviewer and to answer all remarks. Our responses are written in blue in this document.

Finally, some of the figures are too busy and need to be made clearer. For instance, Figure 6 is overly complicated and takes a long time to fully digest.
The Fig 6 and the text have been changed (see your last point).

Also, the "agreement in the dominant contributor" in figure 8 between the models is not clear. How is this agreement determined? What statistical metrics are used? If this is already stated, then please make it clearer!
We agreed it was a missing information. It has been added in the text.
"This rate corresponds to the number of occurrences in the dominant contributor calculated for each hourly concentration in the 4-day forecast over each city. So, a number as 100% over a city shows that both models predict the same dominant country contributor during a 4-day forecast."
And (in bold):
The mean agreement increases up to 75% for determination in the top 5 of the main country contributors to $PM_{10}$ (Fig 11). **In that case, the rate is calculated for the five main country contributors. A score of 100% means both models predict the same five main country contributors for each hourly concentration, but not necessarily in the same order.**"

Minor Comments: P3 L71-73: Provide reference for the WHO health metric stated.
The reference has been added.

P3 L84: Space between "VOCs). The".
Done

P4 L99-101: Please explain in detail how "source" and "receptor" are related in this work to make it clear for readers not familiar with this method.
Additional information has been added (in bold):
"With a such simulation comparison, the simulation with reduced emissions over a source region **(e.g. a country)** allows to highlight the impact of this source on the concentrations over a receptor, **hereafter a city**".

P5 L158: What is the new land-cover dataset used?

The land-cover dataset merges information from GLC-2000 data-set (http://bioval.jrc.ec.europa.eu/products/glc2000/glc2000.php) and CLM database (http://www.cgd.ucar.edu/models/clm/).

The GLC2000 and CLM data-sets through the following procedure:
1. GLC2000 is used to define water, ice, urban and bare surfaces, and then 'high' and 'low' vegetation (HV, LV).
2. Where high vegetation is labelled as sparse, we allocate 50% as HV, 50% as LV.
3. Where low vegetation is labelled as sparse, we allocate 50% as LV, 50% as bare.
4. For each grid square we then allocate the HV and LV vegetation according to CLM categories.

This information is provided in Simpson et al. (2017) – see pages from 116 to 118.

P6 L168-169: Make it clear whether or not other BVOC emissions are used in the model other than isoprene and monoterpenes.

We have added these following sentences:

"The soil-NO emissions of seminatural ecosystems are specified as a function of the N-deposition and temperature (Simpson et al., 2012). The biogenic DMS emissions are calculated dynamically during the model calculation and vary with the meteorological conditions (Simpson et al., 2016)."

P6 L170: The definition of the "remainder" is unclear.

It has been changed. Now it reads:

"… the rest of primary PM defined as the remainder".

P7 L212-213: What do the authors mean by "we have harmonized the used of different parameters"? Do you mean that the setup and input/outputs of the model are been made as consistent as possible?

The following information has been added (in bold):

"To perform properly the analysis between both models, we have harmonized the use of different parameters **such as the horizontal resolution, the anthropogenic emissions used, the definition of the city area and meteorological data used** (Tab. 1)."

P8 L 222: Worth saying that the ECMWF operational system does not archive 3D precipitation fields when this is first discussed on P6.

The information has been added (in bold):

"An estimation of this 3D precipitation can be calculated by EMEP if this parameter is missing in the meteorological fields **as in the data used in this work (see Section 2.4)**."

P8 L 247: Can the authors please elaborate on what they mean by "medium intensity".

The information "no more than three consecutive days beyond the WHO $PM_{10}$ threshold" has been added to the sentence.

P9 L256: What quantifies as "large concentrations"?

It has been added:

"Large concentrations **(>60 μg/m$^3$)** were also predicted…"

P10 L277: I suggest the authors change the word "enormous" to something more scientific.

It has been changed (in bold):

"...to calculate **very large** concentrations **(e.g. hourly concentration higher than 200 μg/m³)**…"

P11: L307-312: I suggest the authors re-write this paragraph as it is unclear and difficult to follow.

It has been re-written. The main corrections are highlighted in bold:

"LOTOS-EUROS **is less correlated** with the concentrations measured by the rural stations than EMEP (Fig. 4). However, **as EMEP,** LOTOS-EUROS also presents **a lower bias with these rural stations in comparison with the urban stations. This** is predictable since with such resolution, the model calculates mainly the urban background concentrations. By comparing the 5 cities having urban and rural stations, as done with EMEP, only the bias and the FGE between the predictions and the **urban** measurements are improved (Fig S4). It is also worth noting that the concentrations predicted by LOTOS-EUROS over these 5 cities are lower than **the ones calculated by** the EMEP model **(in Fig. S3).**"

P11: The discussion of the different metrics is a bit over-kill here. If all this discussion is to be kept in the manuscript, can the authors at least specify what the numbers mean in terms of model performance (e.g. R=0.72 is reasonable and R=0.25 is poor).

This information has been added in Section "3.2.1 Methodology".

"By knowing this point, we have stated that a comparison with the observations presenting for example a correlation coefficient equal to 0.5 or NMB lower than 15% are reasonable results ($r \geq 0.7$ and NMB $\leq 10\%$ are good results)."

P13 L365-371: As mentioned above, I think the authors need to discuss in more detail the source-receptor methodology to make it clear to non-experts of this approach.

It has been rewritten. The changes are highlighted in bold:

"There are in total 31 runs **for each date with reduced anthropogenic emissions. Each run** corresponds to the perturbations for **one of the** 28 countries **related to the 28 EU capitals**, plus Iceland, Norway and Switzerland, giving the contribution for each country.

To calculate the concentration of the pollutant integrated over the studied area, i.e. a selected city, coming from a source, we follow the equation (5):

$$C_{source} = \frac{C_{reference} - C_{pertubation}}{x} \quad (5)$$

With x the reduction in % (i.e. 0.15), $C_{reference}$ is the concentration of the pollutant integrated over the studied area from the reference run and $C_{pertubation}$ is the concentration of the pollutant integrated over the studied area from the perturbation run. **Thus, by differentiating over the studied area, the concentration from the perturbated run with the concentration provided by the reference run, we have an estimation of the influence of the source (i.e. country). By scaling with the reduction used (parameter x), it gives the estimated concentration related to the source**."

P14 L415: "For the positive correlation, a clear feature appears" is an example of these short sentences which break the flow of the text.

We have supposed that the reviewer wanted to quote "For the positive concentrations, a clear feature appears".

It has been deleted and replaced by the part in bold:

"The main contributors to the "Domestic" PM$_{10}$ are POM (~20%) and rest PPM (~30%) (which corresponds to the remainder of coarse and fine PPM), **as noticed for the positive concentrations** (Fig. 6a)."

P15 Section 5.2: If the LOTUS model is using a different approach to that of EMEP, how are the emissions perturbed? This is not overly clear from the text as it stands.
As explained in the introduction, LOTOS-EUROS does not use an emission perturbation scenario but a labelling technique. Thus, the model traces the pollutants through conserved atoms (C, N, S) related to emission sources.
This technique is described in Section 4.2.

P16 L458: What do the authors mean by "each at the end of the EMEP model"?
We have supposed the reviewer wanted to quote "at the opposite of the EMEP model".
We agreed that this sentence was confusing. It has been changed as below:
"As reminder, the EMEP model predicted a slightly larger influence from the "30 European countries" (35%) than from "Others" (25%)."

P16 L459-462: The term "Rest" appears to represent the difference between the total PM and the sum of all its components. Is this the metric used to explain the "non-linearity in the chemistry? If so or if not, I think this sentence need to be rewritten to clear emphasis the definition of "Rest".
The sentence has been changed as:
"In the list of LOTOS-EUROS PM$_{10}$ components there is one named "Rest". "Rest" corresponds to the difference between the total PM$_{10}$ and the sum of all the components, and Fig. 8 shows that it is also a large component of this "Domestic" PM$_{10}$".

P16 L467-8: Is this true? In section 3 I got the impression there was substantial disagreement between the models.
That is certain that both models underestimate the larger peaks observed over the cities. However, both models agree between their predictions.
The reader must remind that the predictions from both models are representative for a large area and will obviously underestimate the concentrations and the contributions for the larger peaks measured by a specific station.
Thus, we have added this sentence in Section 6:
"It has also been shown in Section 3 that both models are representative for a large area and the predictions can underestimate the concentrations and the contributions for the larger concentrations measured by a specific station."
And in the conclusion:
"It may suggest that the both models, which calculate the country contributions over the cities, defined by a large area, may underestimate the contribution measured by a specific station for the higher concentrations."

Figure 2: Could the country outlines be more clearly plotted.
It has been changed.

Figure 6: There is a lot of stuff is this plot, so could be good to make it simpler or bigger at least so easier to see everything. The calculation of non-linearity need to be explained more clearly in the manuscript.
The figure has been split into two parts. Moreover, an explanation in the text has been added:

"This non-linearity has been calculated for each hourly concentration as the standard deviation of the hourly contribution (which can be positive or negative) obtained by the three reduced emissions scenarios and weighted by the hourly total concentration by following the equation (6):

$$NONLIN_{Contrib} = \frac{\sqrt{\frac{\sum_{i=1}^{n}(Ccontrib_i - \overline{Ccontrib})^2}{n}}}{Ctot} \times 100\% \qquad (6)$$

n corresponds to the number of perturbations used (n=3), Ccontrib is the hourly $PM_{10}$ concentration for a specific contribution ("Domestic" or "30 European countries" or "Others") and Ctot is the hourly $PM_{10}$ concentration."

---

## Author Comment (AC3) · 22 Sep 2019

We would like to thank the Editor David Ham to highlight these issues and we would like to apologize for our previous lack.
We have corrected the missing points. Please find our answers written in blue.

I am writing as executive editor to bring to your attention several ways in which this manuscript currently does not comply with GMD policy. These issues will need to be addressed before any revised manuscript could be accepted for publication.

Title
As per https://www.geoscientific-model-development.net/about/manuscript_types. html#item4, the names and version numbers of the models being evaluated need to be included in the title of the manuscript. Please change the title accordingly.
It has been corrected. The title is now:
"Prediction of source contributions to urban background $PM_{10}$ concentrations in European cities: a case study for an episode in December 2016 by using EMEP/MSC-W rv4.15 and LOTOS-EUROS v2.0 - Part.1 The country contributions"

EMEP code on GitHub
GitHub is an excellent development platform, but it is not an archival location suitable for the code used in a paper. Indeed, even GitHub themselves tell you to use Zenodo (https://guides.github.com/activities/citable-code/). Please therefore archive (probably using Zenodo) the precise version of EMEP used in this manuscript, and cite it from the code and data availability section. The Zenodo archive will give you the entry to paste into BibTeX or another reference manager.
LOTOS-EUROS code archive
LOTUS-EUROS causes more issues because the code is not open source. It claims to be open source, however reading its licence indicates that this is not actually true because redistribution is prohibited (see https://opensource.org/osd item 1). This means that properly archiving the version of LOTUS-EUROS used will not be possible. Instead, GMD policy requires you to point out that LOTUS-EUROS is only available under a restricted licence. It remains critical that the manuscript identifies the exact version used in order to enable the results to be reproduced. The manuscript should also not make the incorrect claim that the software is open source. It is unfortunate that this claim is made on the website, however that is not a reason to reproduce the error in the manuscript.
Data availability is missing
This manuscript describes a model evaluation campaign. The models were driven using data, and evaluated using data. The code and data availability section needs to point the user at the persistent public archives for the precisely identified code that was used. For a model evaluation paper, this is likely to make this section quite expansive, in contrast with its current brief extent.
Configuration files, run scripts and evaluation scripts
Reproducibility also demands that the exact configuration files and scripts used to run the models are presented, along the scripts used to process and evaluate the model output. Please also archive and cite this data.
For a fuller description of GMD's code and data evaluation policy, please see: https: //opensource.org/osd

It has been corrected. Now it reads:
"The EMEP model is an open source model available on https://doi.org/10.5281/zenodo.3355041. The base-code of LOTOS-EUROS is available under

the license on https://lotos-euros.tno.nl/, but the code used for this study, including the source apportionment is only available in cooperation with TNO. The data processing and analysis scripts are available upon request."

---

## Author Comment (AC4) · 22 Sep 2019

We would like to thank the reviewer 2 for the helpful comments and corrections. We have answered the different points by highlighting our comments in blue.

This paper compares two source apportionment methods. The methods are not clearly explained. Some clarifications are needed and there are some methodological flaws. Also the English used in this paper needs to be revised.
Some parts of the introduction have been rewritten.

Specific comments: 1. About the comparison between measured and modeled concentration. I understand that the author wants the compare the average concentration over an urban area. From model results it is easy to obtain this, averaging concentrations over some grid cells. Unfortunately you cannot obtain a comparable number from measurements. The stations are not equally distributed over the area of interest and the number of urban, rural and traffic stations might be different. Comparing an average of the stations with an average of the grid cells will introduce an additional uncertainty. Why not interpolate the model results at the station locations and compare with the measurements. A separate comparison for different station types should be made. I think the analysis for one cell, 9 cells and the GADM. I would restrict the analysis to stations inside the GADM.
The comparison by interpolating the model grid cells to the stations present similar results as shown in the paper. The following examples present such comparison and they are comparable to Figs 3 & 4 in the ACPD manuscript.

[Figure]

[Figure]

**Fig.** Scatterplot between the hourly PM$_{10}$ concentrations in µg/m$^3$ over all the studied cities using the 9 grid cells definition, predicted by the EMEP model (top), LOTOS-EUROS (bottom) on December 06[th] 2016 and the observations of the urban sites (blue dot) and rural sites (red square). The EMEP predictions are interpolated to the observations. The four panels correspond to the different predictions from 3 days before the December 06[th] to the actual day, i.e. December 06[th]. The correlation coefficient (r), the mean bias (MB), the normalized mean bias (NMB), the root-mean-square error (RMSE) and the fractional gross error (FGE) are provided on each panel.

However, such comparison does not answer the question about the reliability of our predictions over the cities since the objective was to compare the average concentration and thus the average contribution over each city domain.
We also agreed with a limited number of stations per city, the comparison remains difficult and a regional model will never predict the same concentrations than these sparse measurements.

We have also decided to present the evaluation of the prediction over the cities for the 3 definitions, since we aimed to test different definitions in our products. We also wanted to highlight the importance of the definition of the city boundaries to determine the country contribution.

2. The non-linearity discussed Line 374 and following.
The contributions of individual countries don't have to theoretically sum up to the contribution of all countries reduced together. Even for small reductions there is some non-linearity. But the non-linearity is small for small reductions. The difference between the sum of individual contributions and the joint contribution can be positive or negative. I would not speak about

negative concentrations. You scale up to 100% but in fact you do a source apportionment of the top 15% of the PM10 column. That's perfect and useful for policy. Achieving small emission reductions is already hard enough.

It is an interesting comment from the reviewer. We have however preferred to keep the negative concentrations, since these concentrations highlight the compounds involved in the non-linearity (NO3, H2O and NH4).

3. Validation versus measurements.

The validation shows quite big differences between model and measurements. What is the impact of this error on the source apportionment? To which extent can it be trusted? In regard of this error, which differences between the two methodologies are significant? How certain is it that the biggest contribution is really the biggest?

That is certain that both models underestimate the larger peaks observed over the cities. However, both models agree between their predictions.

The reader must remind that the predictions from both models are representative for a large area and will obviously underestimate the concentrations and the contributions for the larger peaks measured by a specific station.

Thus, we have added this sentence in Section 6:

"It has also been shown in Section 3 that both models are representative for a large area and the predictions can underestimate the concentrations and the contributions for the larger concentrations measured by a specific station."

And in the conclusion:

"It may suggest that the both models, which calculate the country contributions over the cities, defined by a large area, may underestimate the contribution measured by a specific station for the higher concentrations."

4. Figure 6 Maybe it is more useful to present the analysis for some selected cities (and the others in Annex) than for all cities together. The behavior can be quite different across Europe. If non-linearity is small plots for one reduction percentage are sufficient. It is not clear to me which runs were done to obtain these plots.

We have decided to keep the overall description in Fig. 6.

However, the part describing the non-linearity (black horizontal bars) has been shown in another figure (now Fig. 7).

By providing a figure as Fig. 6 for each city will add complexity.

It is right that the impact of non-linearity is not similar for each city. Thus, we have decided to add an additional figure in the supplement with the following text, showing this non-linearity over each city.

[Figure]

**Figure. S10** Mean hourly non-linearity in percent calculated for the "Domestic", "30 European countries" and "Others" contributions, over the 34 European cities and for all 4-day forecasts (i.e. from 01-04 Dec to 09-12 Dec 2016). The non-linearity is presented for the cities defined by 1 grid cell (left row), 9 grid cells (middle row) and by the GADM (right row).

In Section 5.1:
"The mean non-linearity is not homogenously distributed over all cities as shown in Figure S10 and may vary from date to date (not shown). It has remained limited even if some hourly contributions show higher non-linearity. In maximum, 3% of the calculated hourly contributions for all 4-day forecasts over the selected cities have a non-linearity higher than 5% (not shown)."

Indeed, even if some hourly non-linearities may present larger values, the amount of these large non-linearities is limited as shows with this distribution for the 9 grid cells definition:

[Figure]

Figure: Distribution of the non-linearity for the "Domestic", "30 European countries" and "Others" contribution. The line that divides the boxes into two parts represents the median of the data. The end of the boxes shows the upper and lower quartiles. The extreme lines show the highest and lowest value excluding outliers which are represented by grey diamonds (almost seen as a line). The red dots correspond to the mean of each data set.

We have also added these sentences in the conclusion:
"Even if this non-linearity is not identical for all cities and for the different dates, the larger non-linearities (>5%) impact only 3% of all the calculated hourly contributions. However, the non-linearity related to the reduction of each emission precursor has not been calculated in the study for computational reason."

Concerning the runs used for this figure, we have added an additional information in the text:
"This figure is a result of the perturbation runs by separating the positive and the negative concentrations obtained in the calculations. The concentrations have also been gathered by their calculated origin".

On line 358 is mentioned that emission per country where reduced with 15%. Are precursors reduced one by one or all together?
All the anthropogenic emissions are reduced simultaneously. It was explained in the following sentences (lines 361-363). However, we have added the word "anthropogenic" since it was missing.
"The perturbation runs are done for **anthropogenic** emissions of CO, $SO_x$, $NO_x$, $NH_3$, NMVOC and PPM (primary particulate matter). For computational efficiency, in the perturbation calculations, all anthropogenic emissions in the perturbation runs have been reduced here simultaneously."

As also mentioned previously, there is now this sentence in the conclusion:
"However, the non-linearity related to the reduction of each emission precursor has not been calculated in the study for computational reason."

How is the non-linearity calculated? Is it calculated as a share of the total concentration (Line 506). In my opinion it is more correct to use the concentration change as reference?
In the section 5.1, we have added an explanation about the calculation in the non-linearity:

"This non-linearity has been calculated for each hourly concentration as the standard deviation of the hourly contribution (which can be positive or negative) obtained by the three reduced emissions scenarios and weighted by the hourly total concentration by following the equation (6):

$$NONLIN_{Contrib} = \frac{\sqrt{\frac{\sum_{i=1}^{n}(Ccontrib_i - \overline{Ccontrib})^2}{n}}}{Ctot} \times 100\% \qquad (6)$$

n corresponds to the number of perturbations used (n=3), Ccontrib is the hourly PM$_{10}$ concentration for a specific contribution ("Domestic" or "30 European countries" or "Others") and Ctot is the hourly PM$_{10}$ concentration."

It is important to remind, that our calculated contribution (Eq. 5), corresponds to the change in concentrations related to the change in emissions.

5. Comparability of the two methodologies. For primary pollutants both source apportionment methodologies are comparable. Differences are due to differences in the models (transport, deposition,..). But for secondary PM the methods don't necessarily give the same result. E.g. an amount of NOx emitted somewhere can result in a certain ammonium nitrate concentration in the receptor. If NOx is emitted in excess (ammonia limited regime) an emission reduction will have little effect at the receptor point. On the other hand, in the NOx limited regime the same NOx reduction will have a big impact. The labeling method will give the same result in both cases while the 'perturbations' method will give different results. Hence, comparing contributions calculated by the two models is not very useful. The statement on line 513 is not complete: differences are not only due to differences in aerosol chemistry between the models. The reviewer raises an important point.
The following sentences have been added in Section 6:
"It is also related to the differences in both methodologies (e.g. Clappier et al, 2017b). Indeed, an emission reduction and a labelling technique will not necessarily provide the same results for the secondary PM. An emission reduction depends on the atmospheric composition already present. For example, an amount of NO$_x$ emitted over a source can result in a certain NH$_4$NO$_3$ concentration in the receptor. If this NO$_x$ is emitted in excess (NH$_3$ limited regime), a NO$_x$ emission reduction will have a small effect at the receptor point. On the other hand, in the NO$_x$ limited regime, the same NO$_x$ reduction will have a large impact. The labelling method will give the same result in both cases while the scenario approach will give different results."

6. Figure 8 How is the percentage of agreement defined? I think it's more useful to present this for individual cities.
We agreed it was a missing information. It has been added in the text.
"This rate corresponds to the number of occurrences in the dominant contributor calculated for each hourly concentration in the 4-day forecast over each city. So, a number as 100% over a city shows that both models predict the same dominant country contributor during a 4-day forecast."
And (in bold):
The mean agreement increases up to 75% for determination in the top 5 of the main country contributors to PM$_{10}$ (Fig 11). **In that case, the rate is calculated for the five main country contributors. A score of 100% means both models predict the same five main country contributors for each hourly concentration, but not necessarily in the same order.**

In Fig. 8 we have decided to show the mean agreement since to present 34 figures will be unreadable. However, the agreement was calculated for each other compound, as shown below:

[Figure]

[Figure]

**Fig**. Agreement in the determination of the dominant country contributor for $SO_4$, $NO_3$, $NH_4$, EC and POM in percent, and for each 4-day forecast (01-04 Dec 2016, 02-05 Dec 2016, 03-06 Dec 2016, 04-07 Dec 2016, 05-08 Dec 2016, 06-09 Dec 2016, 07-10 Dec 2016, 08-11 Dec 2016, 09-12 Dec 2016) over all the cities using the 9 grid cells definition.

Technical comments:

Line 30: change '15% factor' > '15% emission reduction'. This 15% is not a factor. I think it's confusing. Change this in the whole paper.
It has been changed.

Line 3033: revise grammar, sentence too long. We found that the combination of a 15% reduction and a larger domain help to reduce...
It has been corrected.

Line 36: split sentence
Done.

Line 68: crops yields > crop yields
It has been corrected.

Line 71: states > better established/proposed a PM10 limit value

Changed.

Line 77-79: very unclear contradictory sentence. If a pollutant has a short life time it's impact is close to the source and long-range transport doesn't matter much. Is PM10 really so short lived compared to other pollutants (like NO2). The concentration of PM10 is rather uniform compared to the latter.
We have added the following information (in bold):
"Due to the **relative** short atmospheric life time **(from some hours to days),** the variability is impacted by local sources, meteorological conditions affecting dispersion and long-range transport as well as chemical regimes controlling the efficiency of secondary formation."

Line 81: atmospheric processing? > formed by chemical reactions in the atmosphere.
It has been changed as requested.

Line 85: traffic and transport, all traffic is transport
The word "traffic" has been deleted.

Line 86: biomass burning refers to burning wood for heating. It is an anthropogenic source. You mean wild fires?
It has been replaced by "forest fires".

Line 95: Revise grammar and content. With a country source calculation...????
It has been changed.
"A country source calculation allows to tackle the emissions from the countries responsible for the air pollution episode."

Line 98: revise. Something like: The EMEP calculations use reductions of anthropogenic emissions...
Now it reads:
"The EMEP calculations use reduced anthropogenic emission scenario and compare to a reference run where no changes are applied."

Line 115: Both models are part of...
It has been changed.

Line 122: Use a consistent terminology. You say 'SR system' and the next paragraph is called 'SA system'. A source receptor (SR) model is not a source apportionment (SA) system. Check this through the whole paper. Line 123 and 126: SR? I think it you mean source apportionment. Line 130: be consistent. SR product (should be source apportionment product)... real-time source allocation (=? source apportionment)
Thank you to notice this error.
To be consistent, we have used the term of "source contribution" as presented on the website.
https://policy.atmosphere.copernicus.eu/SourceContribution.php

Line 132: for the 28 EU capitals, plus Bern, Oslo and R.
Changed.

Line 145: too long, split up
Done.

Line 153: ...but the model has also been used...
It has been changed.

Line 161: sigma coordinates? There are... > grammatically incorrect sentence
It has been replaced by:
"The PBL is located within approximately the 10 lowest model levels…"

Line 212: word order! ...cover a slightly different domain...
Corrected.

Line 119: ...by the IFS
"the" has been added.
Since it was the first time that IFS was defined in this paper, we kept the definition and now it reads: "by the Integrated Forecasting System (IFS) of ECMWF".

Line 236: 1 (grid) cell ... 9 (grid) cell... There is only one grid.
It has been corrected.

Line 237: The latter...
It has been changed.

Line 238: ... living area... better 'urban area' or 'build up area'
It has been changed as "build-up area" as requested.

Line 244: BCs ? boundary conditions?
Yes, it corresponds to "boundary conditions". We forgot to define this abbreviation. It has been added when we explained the BCs used, i.e. at the former line 225.

Line 250: repetition
It has been corrected.

Line 255: until fronts moved in
It has been corrected.

Line 258: metrics
Corrected.

Line 259: To properly estimate
Corrected

Line 267: N is the number of the reference dataset? The number of what? Hours? Days?
We have added the following information (in bold):
"number of the reference data set **(e.g. number of observations)**."

Line 283: grid cell
Changed.

Line 285: city edge > city boundary
Changed.

Line 298: ...smoothed over a large domain... Do you mean smoothed over a grid cell? Mis-interpretation > underestimation. So, the correlation is similar for urban and rural stations but urban stations have a bigger bias. That's because peaks are smoothed out over the full cell.
It has been changed. Now it reads:
"In Figure 3, it is also clear that the EMEP model has difficulties to reproduce the highest concentrations measured by the urban stations which are probably smoothed by the model over **the large grid cells as the ones** defining the cities. **The underestimation in** the largest urban concentrations is highlighted by the comparison with the rural stations."

Line 303: By comparing only the 5... remove the comma
Done.

Line 309: grammatically incorrect
It has been corrected.

Line 312: ...than the ones from the EMEP model
Now it reads:
"…than the ones calculated by the EMEP model".

Line 315: globally > In general
It has been changed.

Line 318: ...at the urban...
It has been changed.

Line 332: negative correlation coefficients? Can you explain this better?
It is tricky to interpret these negative coefficients with such limited number of stations. We have added a sentence:
"The correlation coefficient with the rural stations remains difficult to interpret related to the limited number of stations available."

Line 393: only ... as well as... confusing formulation
"as well as" has been replaced by "and".

Line 401: ...one source area.
Corrected.

Line 445: Averaging out over more cells reduces non-linearity. I would not use the term 'negative concentrations'
It has been changed. Now it reads:
"Averaging out over the larger grids reduces globally the non-linearity."

Line 454: confirms the global feature > shows the same trend (?)
It has been replaced by "general trend".

Line 475: reformulate
Done. Now it reads:
"showing that the mean value in the agreement for both compounds **is reduced** by a few low values"

Line 522: ...probably foresees an underestimation... unclear formulation.
"foresees" has been replaced by "suggests"

---

## Author Comment (AC5) · 22 Sep 2019

We would like to thank Alain Clappier for his comments, and we apologize that our analyse could mislead him to unclear conclusions.
We have answered the remarks below, highlighted in blue.

In this article, the authors compare two different source apportionment methods, both able to evaluate how different emission sources contribute to the formation of PM concentrations. The first method is a scenario approach method. It is implemented using the EMEP model to calculate the impact of the reduction of each individual source. The second method is a labelling approach. It is implemented using the LOTOS-EUROS model to calculate the contribution of different sources tracing the mass of the emitted pollutants throughout the different processes computed by the model.

The authors explain that the two methods are comparable only if the concentrations changes related to the scenario approach are not impacted by the non-linearity: Lines 111 to 114 "This highlights the importance to estimate the reliability of both methodologies in the attribution of sources to PM10 concentrations, e.g. to ensure that the concentrations changes related to the scenario approach are not impacted by the non-linearity and to show that both methodologies present similar results."

I have a first serious concern with the way the authors are testing the linearity using the scenario approach: In their article Thunis P. and Clappier A. (2014) show that the non-linearity between emissions and concentrations can affect the impact of the reduction of each individual emission precursors (the concentration reduction is not proportional to the emission reduction) as well as the impact of the reduction of all the emission precursors (the concentration reduction resulting from the reduction of all the precursors simultaneously is not equal to the sum of the concentration reductions resulting from each individual precursor emission).

To test the linearity the authors performed different simulations with EMEP reducing of 5, 15 and 50% all the precursors simultaneously. They claim that reducing the emissions simultaneously or separately may lead to a slight different results.

Lines 383 to 384: "Furthermore, by reducing the emissions simultaneously or separately may lead to a slight different result in the concentrations, but as mentioned previously, this effect is not addressed in this work for computational reason."

How can they claim that the difference between simultaneous reductions and individual reductions is slight. They did not show any results of such test which quantify this difference. Thunis P. et al (2015) show that the non-linearity resulting from the interactions between the different emission precursors is higher than the non-linearity resulting from different reduction percentages. The test performed by the authors can evaluate only a part of the non-linearity which is most likely not the most important part. This test is clearly not sufficient to evaluate the degree of non-linearity. If I refer to what the authors claim lines 111 to 114, they are unable to ensure that the scenario approach and the labelling approaches will give similar results.

Alain Clappier has highlighted an important point.

We agreed that the word "slight" was inappropriate, especially without to show a comparison. It has been deleted.

As also explained in the paper, to perform a test related to the non-linearity in the reduction of each individual precursor will be too time consuming.
Without to count the 9 reference runs corresponding to each date; it will represent in total 1395 runs: 9 dates * 31 countries (sources) * 5 anthropogenic emissions (CO, SOx, NOx, NH3,

NMVOC and PPM). To perform a complete analysis, it should also be done for the three perturbations, namely 5, 15 and 50%, and it will result to 4185 runs in total.

To perform our study, we already performed 837 4-day runs: 9 dates * 31 countries * 3 perturbations + 9 reference runs.

These numbers of runs do not consider the postprocessing of the simulations over the 34 studied cities.

We have added this sentence in section 4.1.2:

"In total, 847 4-day runs have been performed in this work (9 reference runs, and 9 dates $\times$ 31 countries $\times$ 3 perturbations runs)."

And the following information (in bold) in Section 5.1:

"This also shows that the responses to perturbation runs are robust, **even if only the non-linearity in the chemistry related the perturbation used, and not the one related to the reduction of each emission precursor, has been estimated in this study as mentioned in Section 4.1**."

The sentence from line 111 to 114 which was:

"This highlights the importance to estimate the reliability of both methodologies in the attribution of sources to $PM_{10}$ concentrations, e.g. to ensure that the concentrations changes related to the scenario approach are not impacted by the non-linearity and to show that both methodologies present similar results."

aimed to pinpoint the importance to compare both methodologies in our study.

However, to clarify our point, we have modified it. Now it reads:

"**Even if both methodologies mainly aim to answer two different questions, i.e. the emission control scenarios with the scenario approach and the attribution of concentrations from a source by the labelling technique, it is still useful to estimate the reliability of both methodologies in the estimation of the source contribution to $PM_{10}$ concentrations. For example, it is important to ensure that the non-linearity, related to the perturbation used in the scenario approach, has a limited impact on the calculated contributions** and to show that both methodologies **may** present similar results **in the country source attribution**."

We have also added these sentences in the conclusion:

"Even if this non-linearity is not identical for all cities and for the different dates, the larger non-linearities (>5%) impact only 3% of all the calculated hourly contributions. However, the non-linearity related to the reduction of each emission precursor has not been calculated in the study for computational reason."

I have a second serious concern with the way the authors have interpreted the conclusions of the article of Clappier et al. (2017):

In their article, Clappier et al. (2017) illustrate with simple examples that the scenario approach and the labelling approaches gives similar results only if the concentrations changes related to the scenario approach are not impacted by the non-linearity for any kind of percentage reductions from 0 to 100%. This happens for non-reactive species.

Clappier et al. (2017) illustrate also that, even if the scenario approach often shows linearity between emissions and concentrations for a limited reduction fraction (below 50% for example), the results provided by the scenario approach and the labelling approaches are different. That means it is expected that the two methods tested in this article will give different results, even before to start complex simulations.

If I refer again to what the authors claim lines 111 to 114, they should not compare the results of the scenario approach and the labelling approaches because we know they are different. Moreover, comparing different methods using different models ensure with a great certainty that the results will be different. Then, how can we interpret the authors' conclusions? Lines 518 to 519 "It was shown that the results from both source apportionment methodologies agree in average by 68% in the determination the dominant country contributor to the hourly PM10 concentrations and 75% for the top 5 of these country contributors". Are the disagreements shown by the results due to the discrepancy between the methods or to the difference between the models?

As mentioned previously, the sentence in lines 111-114 has been changed. We have also added (in bold), the following information in the abstract:

"Better results are found in the determination the dominant country contributor for the primary component (70% for POM and 80% for EC) than for the secondary inorganic aerosols (50%) **which is predictable due the conceptual differences in the source attribution used by both models**."

And in the introduction:

"Thus, the scenario approach is more appropriate in the calculation of the source contribution for the primary PM components than for non-linear species such as the secondary components (e.g. Burr and Zhang, 2011, Thunis et al., 2019)."

Since the difference in the contributions is mainly seen in the SIA, it shows that it is a clear result related to the difference between both methodologies.

For this reason, we have added this sentence in the conclusion:

"The differences seen are mainly related to the SIA and is a direct consequence of the difference between both methodologies used."

It is also important to note that we have added this following information in the Section 6:

"It is also related to the differences in both methodologies (e.g. Clappier et al, 2017b). Indeed, an emission reduction and a labelling technique will not necessarily provide the same results for the secondary PM. An emission reduction depends on the atmospheric composition already present. For example, an amount of $NO_x$ emitted over a source can result in a certain $NH_4NO_3$ concentration in the receptor. If this $NO_x$ is emitted in excess ($NH_3$ limited regime), a $NO_x$ emission reduction will have a small effect at the receptor point. On the other hand, in the $NO_x$ limited regime, the same $NO_x$ reduction will have a large impact. The labelling method will give the same result in both cases while the scenario approach will give different results."

I have a third serious concern with the way the authors interpret the capacity of the labelling and the scenario approaches to represent the reality:

Lines 386 to 388 the authors mention that: "In their study, Kranenburg et al. (2013) have shown that this technique [the labelling approach] provides more accurate information about the source contributions than using a brute force approach with scenario runs as the chemical regime remains unchanged."

The relation between emissions and concentrations is non linear is the real world as well as in the numerical models. If the results of the scenario approach are changing according to the percentage of reduction and/or the number of reduced emission sources, it is simply because this method is able to reflect reality. Since the reality is non-linear, the scenario approach

method behaves non-linearity. If it is used correctly the method can even quantify the degree of non-linearity.

The labelling approach gives always one unique result, regardless of the degree of non-linearity of the system under study. Because they are not impacted by the nonlinearity, the results are certainly much easier to show. But they give no information about non-linearity showing that the method does not reflect how the system change when the emission change. I fully agree with the authors when they write about the labelling approach: lines 108 to 109, "However, it is not designed to study the impact of emission abatement policies to pollutants concentrations...". It appears clearly that it is nonsense to claim that the labelling approach provides more accurate information about the source contributions than using a brute force approach with scenario runs as the chemical regime remains unchanged.

We have deleted this sentence in our manuscript, which was given in Kranenburg et al. (2013).

As mentioned in our answer to the previous comment, the difference between both methodologies, based on the non-linearity in the "reality" and in the models is explained in Section 6.

To conclude: This article shows significant gaps in the design of the different test as well as in the analysis of the results. I do not understand the usefulness to compare results if it is known in advance that they will be different and if it is know it will be not possible to find the origin of the differences.

Even their respective limitations, we still think that both approaches provide complementary information on source contributions and $PM_{10}$ composition.

The use of both techniques is very useful for quality assurance purposes and our study demonstrates the ability of two modelling approaches to identify source contributions of particulate matter from different countries to several cities in Europe during a pollution episode. The results show a large degree of similarity which is a key result and should be appreciated as there is no way to arrive at the true source apportionment (see FAIRMODE documentation by Mircea et al, in prep)

Thunis, P. and A. Clappier, 2014. Indicators to support the dynamic evaluation of air quality models, Atmos. Environ., 98, 402-409 Thunis P., A. Clappier, E.Pisoni, B.Degraeuwe, 2015: Quantification of non-linearities as a function of time averaging in regional air quality modeling applications, Atmospheric Environment, 103, 263-275. Clappier A., C. Belis, D. Pernigotti and P. Thunis, 2017: Source apportionment and sensitivity analysis: two methodologies with two different purposes. Geosci. Model Dev., 10, 4245-4256.

---

## Editor Decision (ED1)

GMD Topical Editor decision on manuscript, "Prediction of source contributions to urban background PM10 concentrations in European cities: a case study for an episode in December 2016 by using EMEP/MSC-W r4.15 and LOTOS-EUROS v2.0 – Part 1. The country contributions" by Mattieu Pommier et al.

Two nominated referees reviewed the manuscript in June/July, both finding the manuscript required major revisions, also both agreeing to review the revised manuscript.

Two other reviewers made short comments during the discussion phase, each raising some concerns and additional opinions on the methodologies applied, and their limitations.

The authors have made a substantial number of revisions to the manuscript, and have replied to each of the reviewers comments.

Each of the reviewers have assessed the revised manuscript, reviewer 1 recommending acceptance after minor revisions are made, reviewer 2 finding major revisions are still required.

In recommending major revisions, reviewer 2 notes that the authors have addressed many of the reviewer comments, and that "The agreement metric is better explained in this version".

Whilst the reviewer finds "the most problematic ones are not yet answered sufficiently", the only main concern they raise (aside from the wording of some sentences in the findings) is that the moderate model agreement with observed PM10 may compromise the modelling system's calculated country-contributions to the city pollution.

Reviewer 1's minor revisions are restricted to some deficiencies in the way the text is written, with also suggesting the authors could have carried out a more sophisticated analysis.

My opinion, as Topical Editor, is that the manuscript clearly presents the moderate skill of the model, and readers of the paper will then be aware of the level of skill when considering the conclusions and calculated country-associated contributions.

And whilst my Topical Editor decision must be based on the reviewers recommendations, my opinion is that reviewer 2's finding of major revisions is not sufficiently justified, more consistent with minor improvements, albeit with reservations given the only modest model-obs agreement.

Both reviewers identify that the manuscript has improved, and reviewer 2 now seems content with the method description. Given that the level of skill is clearly presented, I agree with reviewer 1, that the paper can be suitable for publication after minor revisions have been made to improve the way some key sentences with the study's findings are communicated.

I have put together a list of minor revisions to sentences in the Abstract and Conclusion, which once addressed, in my view do now make the paper then suitable for publication in GMD.

Topical Editor review

Minor revisions:

1) Title – delete "by" – improves wording of the title

2) Abstract, line 14 – replace "predictable with" with "which is to be expected given"

3) Abstract, lines 22-23 – this sentence "help to reduce the impact of non-linearity…" I am not 100% sure what the authors mean here, but I think they mean there are unresolved non-linearities due to localised sources not included in the model, or processes occurring at finer spatial scales than the model resolves. I therefore suggest to replace "…impact of non-linearity" with "…impact of missing localised sources or sub-grid-scale processes".

4) Abstract lines 23-24 -- Then the next sentence "This non-linearity… is seen" needs to be re-worded – and it's not clear what is meant by "related to the perturbation". I think the authors mean that the missing sources/processes cause a non-linear response not represented in the models. In which case, that needs to be re-worded, suggest to begin something like "That the non-linearity from the fine-scale processes the models are missing is suggested by the nature of the mismatch … ", clarifying how they infer that to be the case. Potentially the sentence "The use of 15% emission reduction…" sentence might be a precursor to strengthen the attribution to missing local-scale processes, so the authors should consider potentially moving that before the "is seen" sentence.

5) Abstract line 30 – The current wording of this sentence makes it difficult for the reader. The subject of the sentence is re: the dominance of the domestic emissions – the sentence can begin "Over the 34 European cities investigated, PM was dominated.." with the wording clarifying to the specific episode better shortened and moved to the end of the sentence, i.e. finish the sentence ".. for the studied episode (December 1$^{st}$ to 9$^{th}$ 2016)."

6) Abstract, line 31 – Reviewer 2 queried this 68% sentence, but I think they mean to clarify the basis on which this 68% agreement is achieved. You have "on an hourly resolution" but I tend to agree with the reviewer 2, the current wording is unclear. I think though, it is a re-ordering of the wording (as in lines 22-24) to put the subject of the sentence first, and have the few words clarifying the nature of the comparison at the end. I mean re-word to something like, "The two modelling systems generally agree on the dominant external country-contributor (68% on an hourly-basis)…".

7) Abstract, final sentence -- Re-word to shorten the final sentence replacing "results are" with "agreement is" and delete "the determination". Similarly, be clearer by re-wording "for the primary component" with "for primary (emitted) species" and similarly "for the secondary inorganic aerosols" with "for the inorganic secondary component of the aerosol".

8) Abstract final sentence -- Also then re: "Better results are found in the determination the dominant…" there is a missing "of" after "determination". But also it needs to be explained what is meant by "predictable". I think you mean you one would expect greater differences due to the compound nature of secondary species. I mean compound in terms of including differences in emissions & removal seen in primary species, but that secondary species also contain additional differences, e.g. between the models re: oxidation and partitioning, that will tend then to inherently make inter-model differences larger for secondary species. An extra sentence could help explain this.

9) Conclusions, last paragraph, 1$^{st}$ sentence – Improve sentence "It was shown that the results from both" with "We show that results from both…."

10) Conclusions, last paragraph, 3rd sentence – again wording needs to be improved here. In the revise text the new sentence (in blue) says "The differences seen are mainly related to the SIA and is a direct consequence of the different between both methodologies used". I think here you're referring to the point at the end of the Abstract re: primary and secondary components, but that sentence added in the revised manuscript needs to be much clearer. Suggest joining this up with the shorter previous sentence, with a shortened version of that, and be clearer if you mean the different methodology in the two modelling systems, or in the method for attributing to the country. I mean have a single sentence instead as something like "Calculating the country attribution on a daily-mean basis has similar agreement, with most differences related to the secondary inorganic component of the aerosol. Best not to use acronym SIA here with it being the main findings.

---

## Author Response (AR2)

Two nominated referees reviewed the manuscript in June/July, both finding the manuscript required major revisions, also both agreeing to review the revised manuscript.

Two other reviewers made short comments during the discussion phase, each raising some concerns and additional opinions on the methodologies applied, and their limitations. The authors have made a substantial number of revisions to the manuscript, and have replied to each of the reviewers comments.

Each of the reviewers have assessed the revised manuscript, reviewer 1 recommending acceptance after minor revisions are made, reviewer 2 finding major revisions are still required. In recommending major revisions, reviewer 2 notes that the authors have addressed many of the reviewer comments, and that "The agreement metric is better explained in this version".

Whilst the reviewer finds "the most problematic ones are not yet answered sufficiently", the only main concern they raise (aside from the wording of some sentences in the findings) is that the moderate model agreement with observed PM10 may compromise the modelling system's calculated country-contributions to the city pollution.

Reviewer 1's minor revisions are restricted to some deficiencies in the way the text is written, with also suggesting the authors could have carried out a more sophisticated analysis.

My opinion, as Topical Editor, is that the manuscript clearly presents the moderate skill of the model, and readers of the paper will then be aware of the level of skill when considering the conclusions and calculated country-associated contributions.

And whilst my Topical Editor decision must be based on the reviewers recommendations, my opinion is that reviewer 2's finding of major revisions is not sufficiently justified, more consistent with minor improvements, albeit with reservations given the only modest model-obs agreement.

Both reviewers identify that the manuscript has improved, and reviewer 2 now seems content with the method description. Given that the level of skill is clearly presented, I agree with reviewer 1, that the paper can be suitable for publication after minor revisions have been made to improve the way some key sentences with the study's findings are communicated.

I have put together a list of minor revisions to sentences in the Abstract and Conclusion, which once addressed, in my view do now make the paper then suitable for publication in GMD.

We would like to thank the Editor for his helpful review. We appreciate his constructive remarks. We have answered the different points by highlighting our responses in red.
To show the changes in the manuscript, we also have written the corrections in red and let the previous corrections (based on the reviewers' comments) in blue.

Topical Editor review
Minor revisions:

1) Title – delete "by" – improves wording of the title
Done

2) Abstract, line 14 – replace "predictable with" with "which is to be expected given"
It has been replaced as requested.

3) Abstract, lines 22-23 – this sentence "help to reduce the impact of non-linearity…" I am not 100% sure what the authors mean here, but I think they mean there are unresolved non-linearities due to localised sources not included in the model, or processes occurring at finer spatial scales than the model resolves. I therefore suggest to replace "…impact of non-linearity" with "…impact of missing localised sources or sub-grid-scale processes".

4) Abstract lines 23-24 -- Then the next sentence "This non-linearity… is seen" needs to be re-worded – and it's not clear what is meant by "related to the perturbation". I think the authors mean that the missing sources/processes cause a non-linear response not represented in the models. In which case, that needs to be re-worded, suggest to begin something like "That the non-linearity from the fine-scale processes the models are missing is suggested by the nature of the mismatch … ", clarifying how they infer that to be the case. Potentially the sentence "The use of 15% emission reduction…" sentence might be a precursor to strengthen the attribution to missing local-scale processes, so the authors should consider potentially moving that before the "is seen" sentence.

The non-linearity refers to the fact that emission changes do not necessarily react linearly to the changes in concentrations. These changes depend on the chemical regime.
To answers both points, (3) and (4), we have changed sentences by (in bold):
 "We found that the combination of a 15% emission reduction and a larger domain (9 grid cells or GADM) help **to preserve the linearity between emission and concentrations changes. The non-linearity, related to the emission reduction scenario used, is suggested by the nature of the mismatch between the total concentration and the sum of the concentrations from different calculated sources**."

5) Abstract line 30 – The current wording of this sentence makes it difficult for the reader. The subject of the sentence is re: the dominance of the domestic emissions – the sentence can begin "Over the 34 European cities investigated, PM was dominated.." with the wording clarifying to the specific episode better shortened and moved to the end of the sentence, i.e. finish the sentence ".. for the studied episode (December 1st to 9th 2016)."

It has been changed. Now it reads: "Over the 34 European cities investigated, $PM_{10}$ was dominated by domestic emissions for the studied episode (December 01st to 09th 2016)."

6) Abstract, line 31 – Reviewer 2 queried this 68% sentence, but I think they mean to clarify the basis on which this 68% agreement is achieved. You have "on an hourly resolution" but I tend to agree with the reviewer 2, the current wording is unclear. I think though, it is a re-ordering of the wording (as in lines 22-24) to put the subject of the sentence first, and have the few words clarifying the nature of the comparison at the end. I mean re-word to something like, "The two modelling systems generally agree on the dominant external country-contributor (68% on an hourly-basis)…".

The sentence is now: "The two models generally agree on the dominant external country-contributor (68% on an hourly-basis) to $PM_{10}$ concentrations."
I prefer to use the word "model" rather than "modelling systems" since both models compose the forecasting system.

7) Abstract, final sentence -- Re-word to shorten the final sentence replacing "results are" with "agreement is" and delete "the determination". Similarly, be clearer by re-wording "for the primary component" with "for primary (emitted) species" and similarly "for the secondary inorganic aerosols" with "for the inorganic secondary component of the aerosol".
It has been corrected as requested.

8) Abstract final sentence -- Also then re: "Better results are found in the determination the dominant…" there is a missing "of" after "determination". But also it needs to be explained what is meant by "predictable". I think you mean you one would expect greater differences due to the compound nature of secondary species. I mean compound in terms of including differences in emissions & removal seen in primary species, but that secondary species also contain additional differences, e.g. between the models re: oxidation and partitioning, that will

tend then to inherently make inter-model differences larger for secondary species. An extra sentence could help explain this.

As mentioned in the previous point (7), the word "determination" has been deleted and "results" replaced by "agreement".

A new sentence has been then added:

"The country contribution calculated by the scenario approach depends on the chemical regime, which largely impacts the secondary components, unlike the calculation using the labelling approach."

9) Conclusions, last paragraph, 1st sentence – Improve sentence "It was shown that the results from both" with "We show that results from both…."

It has been changed as below:

"We have shown that results from both source apportionment methodologies…".

I have used the passive form to be consistent with the rest of the conclusion.

10) Conclusions, last paragraph, 3rd sentence – again wording needs to be improved here. In the revise text the new sentence (in blue) says "The differences seen are mainly related to the SIA and is a direct consequence of the different between both methodologies used". I think here you're referring to the point at the end of the Abstract re: primary and secondary components, but that sentence added in the revised manuscript needs to be much clearer. Suggest joining this up with the shorter previous sentence, with a shortened version of that, and be clearer if you mean the different methodology in the two modelling systems, or in the method for attributing to the country. I mean have a single sentence instead as something like "Calculating the country attribution on a daily-mean basis has similar agreement, with most differences related to the secondary inorganic component of the aerosol. Best not to use acronym SIA here with it being the main findings.

I have corrected the sentence. There are 3 distinct information. Thus, now it reads:

[revised manuscript text omitted]

---

## Author Response (AR3)

Dear Journal and Dr. Graham Mann.

I have corrected the manuscript as requested by Dr. Graham Mann.

These two corrections are highlighted in orange, in addition to the previous corrections (blue to the reviewers and red for the first review given by Dr. Graham Mann).

[revised manuscript text omitted]